# The Role of Drought and Temperature Stress in the Regulation of Flowering Time in Annuals and Perennials

**Min Chen, Tian-Liang Zhang, Chun-Gen Hu and Jin-Zhi Zhang ***

National Key Laboratory for Germplasm Innovation & Utilization of Horticultural Crops, College of Horticulture and Forestry Science, Huazhong Agricultural University, Wuhan 430070, China; minchen970402@webmail.hzau.edu.cn (M.C.); tianliangzhang@webmail.hzau.edu.cn (T.-L.Z.); chungen@mail.hzau.edu.cn (C.-G.H.)

\* Correspondence: jinzhizhang@mail.hzau.edu.cn; Tel.: +86-27-6201-80231; Fax: +86-27-8728-2010

**Abstract:** Plants experience a variety of adverse environments during their vegetative growth and reproductive development, and to ensure that they complete their life cycle successfully, they have evolved specific defense mechanisms to cope with unfavorable environments. Flowering is a vital developmental stage and an important determinant of productivity in the lifetime of plants, which can be vulnerable to multiple abiotic stresses. Exposure to stress during this period can have dramatic effects on flower physiological and morphological development, which may ultimately lead to a substantial loss of yield in seed-producing plants. However, there has been increasing research evidence that diverse abiotic stresses, ranging from drought, low temperature, and heat stress can promote or delay plant flowering. This review focuses on how plants alter developmental direction to balance between survival and productivity under drought and extreme temperature conditions. Starting from the perspective of the functional analysis of key flowering-regulated genes, it is of great help for researchers to quickly gain a deeper understanding of the regulatory effects of abiotic stress on the flowering process, to elucidate the molecular mechanisms, and to improve the regulatory network of abiotic-stress-induced flowering. Additionally, the important agronomic significance of the interaction between abiotic stress and the flowering regulation of perennial plants under climate change conditions is also discussed after summarizing studies on the mechanisms of stress-induced flowering in annual plants. This review aims to clarify the effects of abiotic stresses (mainly drought and temperature) on plant flowering, which are significant for future productivity increase under unfavorable environmental conditions.

**Keywords:** flowering; abiotic stress; drought; temperature; perennial plants





## 1. Introduction

Plants are sessile and cannot move to escape from adverse environmental conditions. Hence, the developmental process of many plants is highly changeable in response to the environmental stresses they encounter. Abiotic stresses, including drought, salinity, devastating temperature (extreme high or low), and nutrient (mainly nitrogen, phosphorus, and potassium) starvation [1], can have a dramatic impact on plant growth and productivity, such as cellular water scarcity, cell membrane damage, enzyme inactivation, and other defects, ultimately leading to severe yield reductions and huge economic losses [2–5]. Therefore, abiotic stress has been an important issue in plant vegetative [6,7] and reproductive [8,9] development, for which the study of abiotic stress effects during reproductive development is of great significance for the maintenance of food production as well as for the world economy.

Flowering is an important agricultural trait in the successful transition of plants from vegetative to reproductive growth, as the optimal flowering time is critical for maximizing reproductive success and ensuring seed production, which is a key step in the evolutionary

success of plants. Due to the continuously fluctuated environmental conditions, plants have evolved specific defense mechanisms to ensure maximum reproductive success [10]. For example, when individuals cannot survive under severe stress conditions, they produce seeds by adjusting the timing of flowering. In recent years, the effects of abiotic stresses on flowering induction have been documented in numerous plant species [11–19]. The ability of abiotic stresses to regulate flowering, with drought and temperature being important stress factors, suggests that plants can combine abiotic stresses effects with flowering signaling pathways. Thus, stress-induced flowering has been recognized as a new means of flowering response due to its important biological benefits throughout the plant life cycle [20].

Extensive physiological and molecular genetic analyses have revealed six major floral regulatory pathways in *Arabidopsis thaliana*, namely, photoperiod, autonomous, thermosensory (ambient temperature perception), vernalization, gibberellin, and age pathways [21–26]. These pathways are ultimately integrated through several flowering genes that regulate flowering in plants. Under long-day conditions, the transcription factor *CONSTANS (CO)* (1, Table 1) acts upstream to activate the expression of *FLOWERING LOCUS T (FT)* (2, Table 1), a core floral integrator gene transcribed in the leaves. Plants sense seasonal changes in day length via leaves, and subsequently, when the optimum environmental conditions are achieved, FT protein is translated into the bast cells as a long-distance signal (florigen), which is eventually delivered to the shoot apical meristem to activate phloem tissue genes, such as *LEAFY (LFY)* (3, Table 1) and *APETALA1 (AP1)* (4, Table 1), to induce flowering in *Arabidopsis* [27,28]. The study of the *Arabidopsis* flower formation pathway is a crucial step in revealing the regulatory network of flower-forming in plants. With the continuation of related research, accumulating evidence indicates that the key integrating genes of the flowering network can also be induced by abiotic stress in regulating flowering time. Abiotic-stress-induced flowering has become a research hotspot, which has attracted extensive attention from worldwide researchers. For example, drought as well as UV-C stress induced the expression of *FT* in *Arabidopsis* [11,29], resulting in early flowering, while the expression of *FT* in *pharbitis* [30,31] was induced by low temperature and nutrient deficiency. Early flowering in response to drought stress in *Arabidopsis* requires the combined function of the flowering gene *GIGANTEA (GI)* (5, Table 1) and the floral integration factor *SUPPRESSOR OF OVEREXPRESSION OF CONSTANS 1 (SOC1)* (6, Table 1) in addition to the upregulation of *FT* [11]. *Arabidopsis* responds to salt stress by inhibiting *FT* and *CO* expression with delayed flowering. Meanwhile, *GI* is also involved in salt-stress tolerance in *Arabidopsis* [32]. These findings suggest that the key integrating genes of the flower-forming network have dual roles in regulating the flowering time and stress tolerance response, and they may be potential genes involved in stress-induced flowering. Taken together, flowering is subject to a combination of an endogenous gene regulatory network as well as external environmental stimuli, which also supports the importance and necessity of further research on abiotic-stress-induced flowering.

Despite its importance, there have been fewer studies concerning the effects of abiotic stresses on plant reproductive development compared to those in the vegetative growth process. Moreover, abiotic-stress-induced flowering has been reported mainly in annual plants, especially during seed germination, seedling growth, and yield. In contrast to annual plants, there are fewer cases of abiotic-stress-induced flowering in perennials, especially woody plants. This may be partly due to the fact that studies involving reproductive development, such as flowering, require longer plant growth cycles and years of study, which can be a great challenge for relevant materials acquisition as well as for researchers. Seasonal flowering is a typical feature that distinguishes perennials from annuals [33]. Citrus is representative of perennial fruit trees that generally bloom once in the spring or several times a year, depending on the variety and genotype, and flowering induction is affected by drought and low-temperature stresses [34,35]. Drought-induced flowering provides an applicable method for shortening the vegetative growth of woody plants, which can be utilized in genetic research and breeding [36]. These cases of stress-induced

vegetative growth and flowering indicate that the vegetative growth and reproductive developmental programs of perennial plants can also be affected by abiotic stress, which might conceal some biological significance [13,36,37]. Therefore, we collected articles on the topic of drought and temperature affecting plant flowering over the last decade through PubMed. The study discusses the similarities and differences between annual and perennial plants. Some insights are provided for further research on the mechanisms by which drought and temperature affect flowering in perennial plants. Accordingly, this review summarizes current knowledge showing that diverse abiotic stresses (mainly drought and temperature) modify the flowering time of plants and examines physiological alterations in their response to stress. Meanwhile, the role of critical flowering regulation genes in response to stress was identified with consideration of the cross-talk molecular mechanisms underlying flowering time regulation and stress response. Finally, valuable agronomic implications of the cross-talk between abiotic stress and flowering regulation under climate change conditions is discussed in perennial plants.

**Table 1.** Specialized terms and their abbreviations appearing in this review.

| Number | Abbreviations | Full-Name | Number | Abbreviations | Full-Name |
|---|---|---|---|---|---|
| 1 | *CO* | *CONSTANS* | 24 | *HDA6* | *HISTONE DEACETYLASE 6* |
| 2 | *FT* | *FLOWERING LOCUS T* | 25 | *FES1* | *FRI ESSENTIAL 1* |
| 3 | *LFY* | *LEAFY* | 26 | *FRL1* | *FRI-LIKE 1* |
| 4 | *AP1* | *APETALA1* | 27 | *FLX* | *FLC EXPRESSOR* |
| 5 | *GI* | *GIGANTEA* | 28 | *SUF4* | *SUPPRESSOR OF FRI 4* |
| 6 | *SOC1* | *SUPPRESSOR OF OVEREXPRESSION OF CONSTANS 1* | 29 | *VRN1* | *VERNALIZATION1* |
| 7 | ROS | reactive oxygen species | 30 | *FUL* | *FRUITFULL* |
| 8 | ABA | abscisic acid | 31 | *VAL* | *CAULIFLOWER* |
| 9 | SD | short-day conditions | 32 | *PIF4* | *PHYTOCHROME-INTERACTING TRANSCRIPTION 4* |
| 10 | LD | long-day conditions | 33 | *MAF2* | *MADSAFFECTING FLOWERING 2* |
| 11 | *TSF* | *TWIN SISTER OF FT* | 34 | *FCA* | *FLOWERING CONTROL LOCUS A* |
| 12 | *SVP* | *SHORT VEGETATIVE PHASE* | 35 | *FVE* | *FLOWERING LOCUS VE* |
| 13 | *FLC* | *FLOWERING LOCUS C* | 36 | *FLM* | *FLOWERING LOCUS M* |
| 14 | *Hd3a* | *HEADING DATE 3a* | 37 | *HOS1* | *HIGH EXPRESSION OF OSMOTICALLY RESPONSIVE GENE 1* |
| 15 | *RFT1* | *RICE FLOWERING LOCUS T1* | 38 | HSR | heat stress response |
| 16 | *Ehd1* | *EARLY HEADING DATE 1* | 39 | HSPs | heat shock proteins |
| 17 | *RCN1* | *RICE CENTRORADIALIS 1* | 40 | HSFs | heat stress transcription factors |
| 18 | *FD* | *FLOWERING LOCUS D* | 41 | *BOB1* | *BOBBER1* |
| 19 | FAC | florigen activation complex | 42 | *FTL3* | *FLOWERING LOCUS T-like 3* |
| 20 | *FRI* | *FLOWERING CONTROL LOCUS A* | 43 | *PRR* | *PSEUDO RESPONSE REGULATOR* |
| 21 | *OST1* | *OPEN STOMATA 1* | 44 | *LUX* | *LUX ARRHYTHMO* |
| 22 | *VOZ1* | *VASCULAR PLANT ONE-ZINC FINGER 1* | 45 | Eps-D1 | Earliness *per se* locus |
| 23 | *RFS* | *Regulator of Flowering and Stress* | 46 | *EG1* | *EXTRA GLUME 1* |

## 2. Drought-Induced Flowering

Global warming and the continued increase in the world's population have led to a shortage of freshwater resources and a further decline in groundwater levels, posing a major challenge to agriculture worldwide [38]. Drought, which is defined as being in a state of water shortage for several consecutive weeks [39], is the most common abiotic stress around the world and severely affects flowering time, flower morphological developmental processes, and the seed productivity of several plant species. It is particularly noteworthy that drought stress can also cause flower abortion and eventually plant sterility by altering the expression levels of various genes critical to flowering regulation pathways, which regulate both flowering time and response to drought stress [40]. In the following, we discuss how plants perceive and respond to drought stress, further summarize the differential physiological phenotypes of various plant species under drought stress conditions, and, finally, we focus on the potential molecular mechanisms underlying drought-stress-induced flowering.

### 2.1. Perception and Coping Strategies of Drought Stress

Plants perceive drought stress signals mainly through the leaves and the root system. Stomatal movement can be observed in leaves, and drought stress can lead to the accumulation of reactive oxygen species (ROS) (7, Table 1) and of abscisic acid (ABA) (8, Table 1) in leaves, which, in turn, regulates the movement of guard cells and ultimately determines the opening and closing state of stomata [41]. However, it is difficult to determine how plants respond to drought stress in the root system [42,43]. A deficiency of water can constrain the growth and development process of plants, and can even have a significant impact on plant survival [1,44]. As a result, plants have evolved a variety of strategies to cope with damage caused by drought stress. The process by which plants sense water deficit signals and further initiate coping strategies in response to drought stress is known as drought resistance. The adaptability of plants to drought stress mainly consists of three different coping strategies, namely, drought escape, drought avoidance, and drought tolerance [45]. Drought escape, a common strategy exploited in response to drought stress, refers to plants that accelerate flowering and shorten their entire life cycle before severe drought stress hinders their survival [46,47]. However, in order to achieve early flowering, a drought escape strategy will terminate vegetative growth in advance, which can severely influence the growth and development of vegetative organs, and eventually lead to a dramatic decrease in seed yield. Drought avoidance (also known as drought dehydration) is another strategy for plants to cope with external drought conditions by increasing the internal water content (by reducing water loss or maximizing water uptake) [48]. The drought tolerance strategy is the ability of plants to tolerate low internal water content and to adapt to the drought stress while initiating reproduction [49].

Under drought stress conditions, plants can respond by early or late flowering, depending on the onset, duration, and severity of drought [20,50]. A bibliometrics analysis showed that plant response to drought has become an important research topic [51]. When plants are adequately supplied with water, the stomata remain open to a large extent to enable the plants to fully photosynthesize, while under mild drought stress, plants will appropriately regulate stomatal closure to minimize water loss by reducing transpiration, but this will result in a decrease in the rate of photosynthesis. When subjected to severe drought stress, the stomata are generally in a minimally open state to ensure that some photosynthesis can take place, thus guaranteeing the normal survival needs of plants [52]. This is one of the main approaches for plants to avoid damage caused by drought stress in the short term [53]. Influenced by geography, many terrestrial plants are frequently affected by drought stress and have developed various drought-tolerant mechanisms to adapt to or to resist the drought environment through a long-term evolutionary process [54,55]. The adaptation of plants to the drought environment is mainly reflected both morphologically and biochemically [56–58]. Morphologically, adaptation is manifest in the presence of a very thick cuticle on the leaf surface, with the fenestrated cells being tightly arranged, while

some leaves have a tomentum on the surface, which can effectively control water loss, and can also absorb dew at night to replenish the plant's own water [59]. Generally, there is a very well-developed root structure with greater water and nutrient absorption capacity and a poorly developed aboveground branching structure with weaker transpiration and better water retention capacity [60]. The more drought-resistant the plant, the greater the root–crown ratio. These morphological adaptations are closely related to the cell division, elongation, and differentiation of the root apex. Plant vascular tissue systems, such as the xylem and phloem, are involved in the transport of substances, while their developmental status also affects plant drought resistance [61]. In *Arabidopsis*, drought-escape-induced early flowering is associated with the phloem tissue transport of the florigen FT protein from the leaves to the shoot apical meristems [62]. Biochemically, it is manifested in the high expression of some drought resistance genes that positively increase the content of amino acids and sugars in plants (such as proline and trehalose), the enhanced activity of antioxidant-related enzymes, and inhibition of the activity of enzymes involved in degradation pathways to ensure that normal metabolic homeostasis is maintained under drought stress condition [63]. ROS, including superoxide radicals ($O_2^-$), hydrogen peroxide ($H_2O_2$), and hydroxyl radicals ($OH^-$), regulate plant growth and development at lower concentrations [64,65]. Excessive accumulation of ROS under drought stress leads to membrane lipid peroxidation [66,67]. Previous studies have shown that excessive ROS in plants will be scavenged by antioxidant mechanisms, including the enzymatic antioxidants, SOD (superoxide dismutase), CAT (catalase), POD (peroxidase), and the non-enzymatic antioxidants, ascorbic acid, proline, flavonoids, and polyphenols, which ultimately improve the plant's drought resistance [68–71]. Ascorbic acid has also been reported to play a role in controlling the flowering time in plants [72]. Understanding the perception of drought stress and the coping strategies (early or late flowering) used by plants in response to drought provides a physiological basis for subsequent studies on the molecular mechanisms of drought-stress-induced flowering.

*2.2. Flowering Time of Various Plant Species in Response to Drought Stress*

Most plants have evolved and adapted to the frequent fluctuations in the natural environment, especially the severe damage caused by droughts due to water deficit. Drought stress can lead to alterations in the flowering time of various plants, effects on flower development (including reduced flower number, restricted filament elongation, and delayed anther development), immature seed development, and reduced yields [40,73,74]. Therefore, the flowering time is an important agricultural characteristic for the development of adaptation to drought stress in a wide range of plants (Table 2). The discrepancy between early and late flowering resulting from the effects of drought stress depends on the plant species [20,75]. For example, forage and biofuel crops generally have delayed flowering as a desirable target for them due to the importance of the plant vegetative biomass. Cereal crops, by contrast, usually exhibit early flowering as an ideal trait to shorten the harvest time, and, thus, increase the number of plantings during the growing season, while growing as fast as possible to minimize damage from drought stress caused by environmental fluctuations [73,76]. Drought stress caused by water deficit delays the flowering time of *Arabidopsis* under short-day (SD) (9, Table 1) conditions but accelerates flowering time under long-day (LD) (10, Table 1) conditions [11,46]. Drought stress causes premature bolting of Chinese cabbage in the growing season, which leads to insufficient vegetative growth and influences yield and quality [77]. By studying the effect of drought stress on flowering time in *Brassica rapa* offspring, it was found that materials with seeds collected after drought flowered earlier than those collected before drought, suggesting that *Brassica rapa* responds to drought stress and evolves towards earlier flowering [78]. Studies of drought-induced flowering in the chickpea suggest that drought stress generally accelerates the flowering time of temperate grain legumes [79,80]. The role of drought in influencing the flowering time varies among plant species and with environmental conditions, so that drought regulation of flowering is the result of multiple factors.

Recently, there have been studies on drought-induced flowering in several other plant species [12,31,81–83]. In contrast to early flowering induced by drought stress, the flowering time of rice is delayed under water deficit conditions to avoid reproductive growth in unfavorable environmental conditions, but water shortage still results in the retardation of plant growth and spikelet development, which leads to reduced crop yield and ultimately economic losses [84–86]. Water deficiency can also delay the first flowering of *Medicago polymorpha*. The effect of drought stress on early and late flowering in plants is closely related to the intensity and duration of the water deficit, in addition to varying by plant species. The artificial control of the duration and intensity of drought treatment in agricultural production plays an important role in accelerating plant development, especially regarding flowering. It was found that the response of rice to drought stress was dependent on the intensity of drought, and mild water deficit in the early development stage triggered a drought escape response with accelerated flowering and reduced tillering [87,88]. In a study of wheat response to drought stress, it was also found that the flowering time showed a nonlinear relationship with the plant water content status, with mild water deficit shortening the flowering time, while severe drought stress delayed flowering [89].

Drought-induced flowering is a phenomenon more common in annual plants. The drought stress regulation of plant flowering has been less well studied in woody plants and remains poorly understood, mainly due to the long duration of vegetative growth in perennials and the excessive long period of research. Drought stress is one of the major environmental factors inducing flowering in adult citrus in subtropical regions. Different citrus species are induced to flower by different environmental conditions, with lemon, four-season orange, and kumquat being mainly affected by drought stress, while sweet orange, trifoliate orange, mandarin orange, grapefruit, and tangerine are mainly affected by seasonal low temperature [39,90]. Notably, drought-induced flowering in citrus was also accompanied by the upregulation of the *CiFT* expression level [37,90]. Earlier studies have also shown that *Citrus latifolia* flowering is also induced by drought stress, and that all citrus plant species share this same flowering mechanism [91,92]. Additionally, the perennial woody plant *Sapium sebiferum* takes 3–5 years to flower normally, but one-year-old seedlings under drought stress flower early, which provides a feasible way to shorten the vegetative growth years of woody plants in genetic research and breeding efforts [36]. These indicate that the effect of drought stress on the flowering time is not specific to a particular plant species but is conserved in annuals as well as perennials. There are few studies on the regulation of flowering time by drought stress in perennials due to the long study time and the lack of phenotypes, but the available evidence supports the feasibility of researching this topic in perennials. Moreover, studies on drought-induced flowering in perennial plants can be carried out on the basis of sufficient theoretical evidence in annual plants.

## 2.3. Molecular Regulatory Mechanisms of Flowering Involved in Drought Stress

Drought-stress-induced flowering, as well as the traditional flower formation pathway, accomplish the same flowering purpose, but the traditional pathway is the primary option for flowering under normal environmental conditions, whereas drought-induced flowering is an emergency response under stressful conditions [20,93]. Compared with the in-depth studies of the flower formation regulatory pathways in plants, the molecular regulatory mechanisms of flowering involved in drought stress are still obscure. Drought stress triggers the differential expression of a variety of genes, including flowering time regulation genes and transcription factors associated with the stress response. Among them, the key genes of flowering regulation in response to drought stress tolerance are *FT*, *CO*, *LFY*, *GI*, *SOC1*, and *TWIN SISTER OF FT (TSF)* (11, Table 1) [11,40,46,94] (Figure 1).

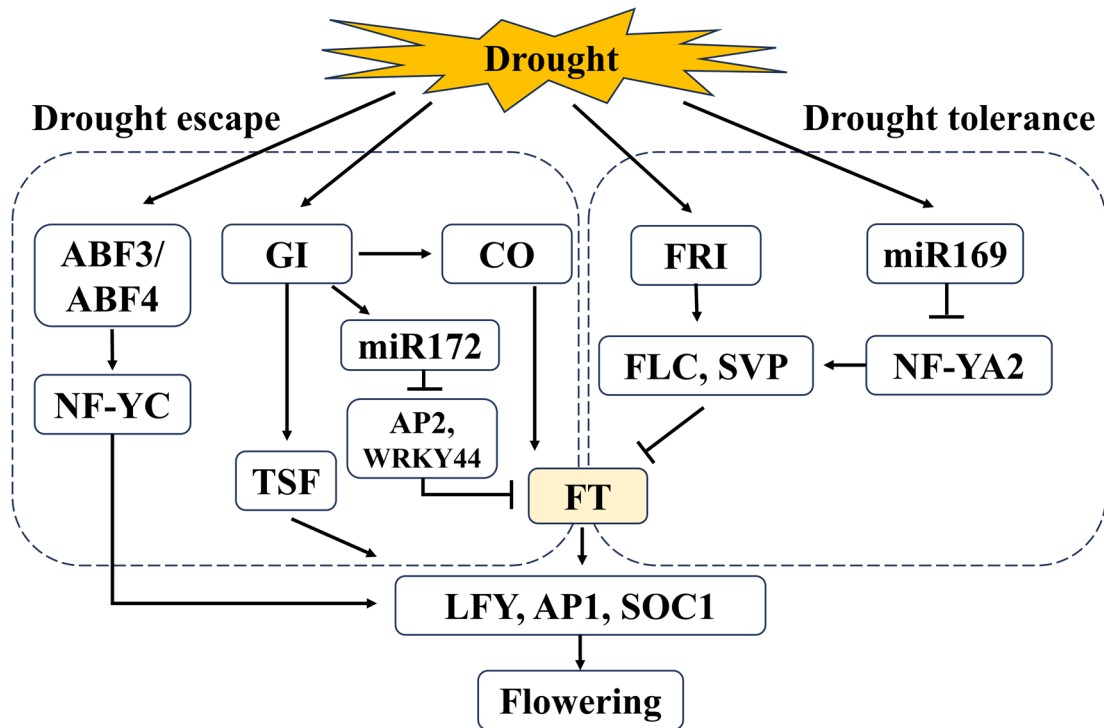

**Figure 1.** Simplified regulatory pathways linking drought stress and flowering in *Arabidopsis thaliana*. Drought escape: *ABF3/ABF4* further activates the expression of *LFY*, *AP1*, and *SOC1* by targeting *NF-YC* [95]. *GI* accelerates flowering under drought conditions by positively regulating the expression of *CO* and *miR172* [94], which, in turn, activate the expression of *FT*, or directly activate the transcription of *TSF*, which ultimately upregulates the expression levels of *LFY*, *AP1*, and *SOC1* [11,32,46]. Drought tolerance: *miR169* targets *NF-YA2* to reduce its transcriptional abundance [96], which attenuates the repressive effect on downstream genes *FLC* and *SVP* [97], while *FRI* positively regulates the expression of *FLC* and *SVP*, resulting in the repression of *FT* transcription and delayed flowering under drought conditions [98]. Solid lines indicate identified associations, arrows indicate positive regulation, and horizontal bars indicate negative regulation.

The molecular mechanisms by which drought stress regulates flowering time in *Arabidopsis* have been partially elucidated. Emerging evidence suggests that *GI*, a photoperiodic pathway gene that promotes flowering, is a pivotal regulator of the abiotic stress response and can influence plant tolerance to abiotic stresses, especially drought [11,99]. Under long-day environmental conditions, water deprivation achieves drought-induced early flowering in *Arabidopsis* through ABA-dependent control of *GI* signaling that activates expression of the florigen genes *FT* and *TSF*. Under short-day conditions, the drought and plant stress hormone ABA is considered to inhibit the transcription of *FT* and *TSF* via activating repressors of floral formation, which, in turn, leads to late flowering in Arabidopsis [11,20,32,46,100]. It has been confirmed that the *GI-miR172* pathway is involved in drought-induced early flowering by downregulating *WRKY44* (directly repressed by miR172) [94] (Figure 1). Several other flowering inhibition genes are also induced by drought stress. For example, water deficit induces the flowering repressor gene *SHORT VEGETATIVE PHASE (SVP)* (12, Table 1), which represses the transcription of genes related to ABA catabolism, and increases ABA accumulation, which improves drought tolerance in *Arabidopsis*, but flowering is delayed [97]. Similarly, *FLOWERING LOCUS C (FLC)* (13, Table 1), a flowering suppressor gene, also plays a role in the drought stress pathway, and the loss of *FLC* function leads to early flowering and decreased drought tolerance in *Arabidopsis* [101]. In rice (*Oryza sativa*), activation of the florigen genes *HEADING DATE 3a (Hd3a)* (14, Table 1), flowering integration factor *OsMADS50* (an orthologue gene of *SOC1* in *Arabidopsis*), and *RICE FLOWERING LOCUS T1 (RFT1)* (15, Table 1) (*AtFT*-like

gene) coordinates the modulation of the drought escape response [87]. Meanwhile, the CCT domain protein Ghd7 plays an important role in delaying the rice heading date and regulating drought stress tolerance under long-day conditions [102,103]. The transcription levels of *Hd3a*, *RFT1*,and *EARLY HEADING DATE 1* (*Ehd1*, upstream of the florigen genes) (16, Table 1) are drastically decreased under drought environmental conditions, which eventually leads to delayed floral transition [85]. *RICE CENTRORADIALIS 1* (*RCN1*, an orthologue of *TFL* in *Arabidopsis*) (17, Table 1) is reported in rice as a flowering time regulation gene in the pathway of drought-regulated floral transition that interacts with the 14-3-3 protein and OsFD1 to repress Hd3a protein function but not its transcriptional level, causing delayed flowering in rice under drought stress [84,104] (Table 2). These results suggest that when plants are subjected to drought stress, a large number of genes are induced to be expressed, including genes critical to the flowering pathway, and that differences in the expression of these genes between species ultimately lead to different flowering outcomes.

**Table 2.** Some examples of stress-induced flowering associated with flowering pathway genes. The yellow areas show examples of flowering induced by drought stress, the blue areas represent low temperature stress, and the red areas represent heat stress.

| Abiotic Stress Factors | Species | Flowering Response | Related Flowering Pathway Genes | References |
|---|---|---|---|---|
| Drought (LD) | *Arabidopsis* | early flowering | *FT, GI, SOC1, TSF* | [11] |
| Drought (SD) | *Arabidopsis* | delayed flowering | *FT, TSF* | [46] |
| Drought | Rice | early flowering | *Hd3a* (*AtFT*), *OsMADS50* (*AtSOC1*), *RFT1, Ehd1, OsTIR1, OsABF2, OsmiR393* | [85,87,105] |
| | | delayed flowering | *RCN1* (*AtFT*) | [84] |
| | Maize | early flowering | *ZmNF-YA3* | [106] |
| | Barley | early flowering | *miR172, AP2-like* | [107] |
| | Citrus | induction | *CiNF-YA1* | [92] |
| | | | *CiFD* | [13] |
| | *Brachypodium* | delayed flowering | *BdRFS* | [17] |
| | *Solanum lycopersicum* | early flowering | *SlOST1, SlVOZ1* | [18] |
| | *Arabidopsis* | delayed flowering | *OXS3, AP1* | [49] |
| | | | *OXS2, SOC1* | [82] |
| | | early flowering | *ABF3/4, NF-YC, SOC1* | [95] |
| | | | *miR169d, AtNF-YA2* | [96] |
| | *Sapium sebiferum* | induction | *GA1, AP2, CYR2* | [36] |
| Low temperature | *Arabidopsis* | delayed flowering | *FCA, FVE, SVP, FLM* | [23,108] |
| | | | *MAF2* | [109,110] |
| | | | *HOS1, CO, FLC* | [111–113] |
| | | | *HOS15, GI* | [114] |
| | | early flowering | *miR169d, AtNF-YA2* | [96] |
| | *Phaibitis* | induction | *PnFT1, PnFT2* | [30,31] |
| | Chrysanthemum | induction | *MAF2* (*AtFLC*) | [110] |
| | Poplar | induction | *FT1* | [115] |
| | Citrus | induction | *CiNF-YA1, CiFT* | [92] |
| | | | *CiFD* | [13] |
| | | | *CiFT, CsFT* | [39,116] |
| | *Medicago sativa* | delayed flowering | *MsFRI-L* | [117] |
| | *Barley/Wheat* | induction | *VRN1, VIN2, VRN3* | [118–120] |
| | *Cymbidium goeringii* | induction | *CgSVP* | [121] |
| Heat stress | *Arabidopsis* | early flowering | *FT* | [122] |
| | | | *PIF4, PIF5* | [123,124] |
| | Soybean | induction | *GmFT2a, GmFT5a* | [125] |
| | Barley | delayed flowering | *FLC* gene family | [126] |
| | Rice | early flowering | *EG1, OsGI* | [127] |
| | Maize | early flowering | *ZmNF-YA3, ZmFTL12* | [106] |
| | Chrysanthemum | delayed flowering | *FTL3* (*AtFT*) | [128] |
| | *Brassica rapa* | delayed flowering | *H2A.Z, FT* | [129] |

A nuclear factor-Y (NF-Y) transcription factor, *ZmNF-YA3*, has the dual function of promoting maize flowering while increasing plant drought tolerance, but there is a lack of evidence on the specific mechanism of *ZmNF-YA3* in drought-affected maize flower formation [106]. Also, *Arabidopsis ABF3* and *ABF4* act with *NF-YCs* to mediate drought-

accelerated flowering by regulating *SOC1* [95] (Figure 1). In citrus, *CiNF-YA1* was also found to promote drought-induced flowering by forming a complex with *CiNF-YB2* and *CiNF-YC2* to activate *CiFT* expression, and overexpression of *CiNF-YA1* in citrus increased plants drought-sensitivity [92]. It is evident that *NF-YAs* are likely to be functionally conserved in regulating flowering in annual and perennial plants, with functional diversity resulting from physiological differences in response to stress. These studies support a critical role for NF-YAs in promoting not only the flowering time but also drought response (tolerance/sensitivity). However, future studies are needed to clarify whether NF-YAs are directly involved in regulating drought-affected flowering. The bZIP transcription factor *FLOWERING LOCUS D (FD)* (18, Table 1), together with FT and the 14-3-3 proteins, is the florigen activation complex (FAC) (19, Table 1) that regulates plant flowering [130]. *CiFD* was found to form two distinct proteins through alternative splicing, *CiFDα* and *CiFDβ*, that both initiate flowering in citrus. Among them, *CiFDα* was induced by low temperature while *CiFDβ* was induced by drought stress. The regulatory mechanism of *CiFDβ* promoting drought-induced flowering is independent of FAC and interacts directly with *AP1* [13] (Table 2). *FRIGIDA (FRI)* (20, Table 1) is an essential regulator of flowering in various plant species, including *Populus balsamifera* [131], *Medicago sativa* [117], *Brassica napus* [132], and *Vitis vinifera* [133]. Importantly, *FRI* modulates drought tolerance through the *FLC–OST1* regulatory module [101] (Figure 1). *CiFRI*, a homologue of *FRI* in citrus, was drought-induced, and overexpression of *CiFRI* enhanced drought tolerance in *Arabidopsis* and citrus, whereas silenced plants showed drought sensitivity, and the ectopic expression in *Arabidopsis* exhibited late flowering. The citrus dehydrogenase gene *CiDHN* may maintain the stability of the CiFRI protein during drought-induced degradation [14]. Therefore, the drought-induced flowering regulation genes are conserved between annual and perennial plants. Together, these studies strongly support the pivotal roles of flowering-time-regulated genes in drought stress response and tolerance. The main challenge in woody plants is that the regulatory role of genes in drought-induced flowering can be demonstrated, but there is no phenotypic evidence of flowering, which is related to its own longer developmental process.

In addition, drought-induced transcription factors are closely related to the existing flowering regulatory pathways, and these TFs affect the flowering process of plants by regulating the transcription level of flowering-regulated genes [12,17,134]. The tomato *OPEN STOMATA 1 (SlOST1)* (21, Table 1) loss-of-function mutant causes reduced drought tolerance in plants, and the *slost1* mutant exhibits a late-flowering phenotype under both normal and drought environmental conditions. *SlOST1* combines with the flowering integrated gene *VASCULAR PLANT ONE-ZINC FINGER 1 (SlVOZ1)* (22, Table 1) to form a regulatory module, and then interacts with the promoter of *SINGLE FLOWER TRUSS* to regulate tomato flowering under drought stress [18]. A conserved and specific gene family in plants, the *Regulator of Flowering and Stress (RFS)* family (23, Table 1), produces dramatic alterations in transcriptional levels in response to drought environmental stimuli. Overexpression of *BdRFS* in *Brachypodium distachyon* not only substantially delayed flowering but also promoted drought tolerance. The *rfs* mutants in *Arabidopsis* and *Brachypodium distachyon* displayed an early flowering phenotype and were susceptible to water deprivation [17].

Studies have reported that epigenetic mechanisms, including histone acetylation as well as methylation, are involved in the plant stress response and flowering time regulation. Histone deacetylase *HISTONE DEACETYLASE 6 (HDA6)*-deficient mutant plants (24, Table 1) exhibited a phenotype of reduced drought stress tolerance and delayed flowering with the repression of *FLC* expression [135–137]. The histone H4 gene *BrHIS4.A04*, which interacts with *BrVIN3.1*, is overexpressed in Chinese cabbage and reduces plant susceptibility to drought stress and accelerates flowering under normal growth conditions, whereas under water deficit environmental conditions, the histone H4 gene represses the expression of photoperiodic flowering genes to prevent premature bolting [77]. The regulation of flowering time under drought stress is also related to microRNAs (miRNAs). miRNAs are considered to be important suppressors of gene expression at the transcriptional and post-

transcriptional levels. The involvement of miRNAs in regulating drought-stress-induced plant flowering responses has been found in many species, such as the annual plants *Arabidopsis* [138], rice [139], wheat [140], and maize [141], as well as in perennial plant species [142]. miR172 acts in the process of drought tolerance and flowering time regulation. miR172b-3p and miR172b-5p, derived from a common precursor, promoted flowering and enhanced drought tolerance in barley (Table 2). miR172b-3p expression was upregulated under drought stress treatment, which suppressed the four *AP2*-like transcription factors in barley to accelerate flowering. The expression of miR172b-5p was inhibited under drought conditions; thus, trehalose-6-phosphate synthase (TPS), a key enzyme for trehalose biosynthesis targeted by miR172b-5p, was significantly accumulated to enhance drought tolerance in barley [107]. miR156, which is in the same age pathway regulating plant flowering as miR172, is also induced by drought stress and delays flowering of *Arabidopsis* and tobacco [138,143]. miR169 family members play an important role in stress-induced flowering by inhibiting *NF-YA2*, which, in turn, decreases *FLC* expression, allowing the promotion of flowering [96] (Figure 1). In OsmiR393-overexpressing rice plants, miR393 responds to drought stress by targeting and, thus, repressing the expression of the auxin receptor genes *OsTIR1* and *OsAFB2* for early flowering [105]. In addition, differential expression of key proteins and post-translational modifications, such as SUMOylation, act in regulating plants' flowering processes under drought stress conditions [144]. Taken together, when plants are subjected to drought, a variety of molecular regulatory mechanisms can be activated, suppressed, and integrated to maintain survival through the adjustment of flowering time.

## 3. Temperature-Induced Flowering

With global climate fluctuation, extreme temperature conditions become more intense and more frequent, and temperature becomes another abiotic stress factor that has an enormous impact on flowering time [145]. Temperature stress, including heat and cold stress, are a serious threat to the physiological and developmental processes of plants, particularly in terms of floral transition and crop productivity [20,93,146]. For many plants, the process of prolonged exposure to non-freezing temperatures to allow for a successful floral transition the following spring is called vernalization. Heat and cold acclimation have been considered as a possible strategy for plants to mitigate the damage caused by temperature stress [12]. Clearly, understanding the response of plants to diverse environmental temperatures is essential to discriminate between the effects of vernalization and temperature stress on plants' floral transition. Below, we discuss the vernalization pathway in model plants, further summarize the differential physiological phenotypes of various plant species under cold and heat stress conditions, and, finally, we focus on the potential molecular mechanisms underlying temperature-induced flowering.

### 3.1. Vernalization-Mediated Floral Transition

The successful transition of plants from vegetative growth to reproductive development relies on the completion of flowering at an appropriate time to cope with the hazards caused by adverse environmental conditions. Accordingly, several plant species have evolved mechanisms to integrate diverse environmental cues, thereby coordinating flowering time with favorable seasonal conditions, and the vernalization pathway constitutes a typical example of such a process [147–149]. Vernalization is an essential adaptation of plants to natural environmental temperatures, ensuring the acquisition of the competence to flower in spring by prolonged exposure to low winter temperatures so that reproductive developmental processes can be carried out under appropriate temperature conditions [150]. This programmed physiological response does not directly initiate flowering, but rather provides meristematic tissue with the ability to perceive environmental flowering signals. In temperate regions, many annual winter crops, such as wheat (*Triticum aestivum*) and barley (*Hordeum vulgare*), flower in the spring of the following year, which is conducive to successful reproduction, and adequate exposure to low temperatures for

several months in winter (0–10 °C for about a month or more) is one of the determinants of their flowering [151–153]. Vernalization is also widely found in biennial and perennial plant species, such as vegetables and fruit trees, which require a prolonged period of cold to break dormancy and initiate flowering [154]. Plant perception of low temperature during the vernalization response consists of two distinct but continuous processes, namely, cold perception to induce tolerance and output of a vernalization dosage to accelerate the developmental transition [154,155].

The major molecular mechanisms and diverse genetic networks controlling vernalization have been intensively studied in several plant species, especially in *Arabidopsis*, wheat, and barley. In the model plant *Arabidopsis*, the vernalization requirement is, to a large extent, conferred by the interaction of two determinant proteins: the MADS-box protein FLC [156,157] and the scaffold protein FRI [158]. In the autumn prior to winter cold exposure, *FRI* transcriptionally activates *FLC* by forming complexes with *FRI ESSENTIAL 1 (FES1), FRI-LIKE 1 (FRL1), FLC EXPRESSOR (FLX),* and *SUPPRESSOR OF FRI 4 (SUF4)* (25–28, Table 1) [98]. High levels of *FLC* expression subsequently act both in leaves and meristems to delay flowering through the transcriptional repression of genes encoding the flowering promoters, such as *FT, FD, SOC1,* and *LEAFY* [159,160] (Figure 2). Therefore, winter-annual *Arabidopsis* undergoes vegetative growth in the fall, and the vernalization response leads to the inhibition of the floral repressor *FLC* expression, which ultimately initiates flowering in the following spring [161]. In contrast, *Arabidopsis* summer annuals with lower *FLC* expression can flower rapidly within a single growing period, thus completing the entire life cycle [98].

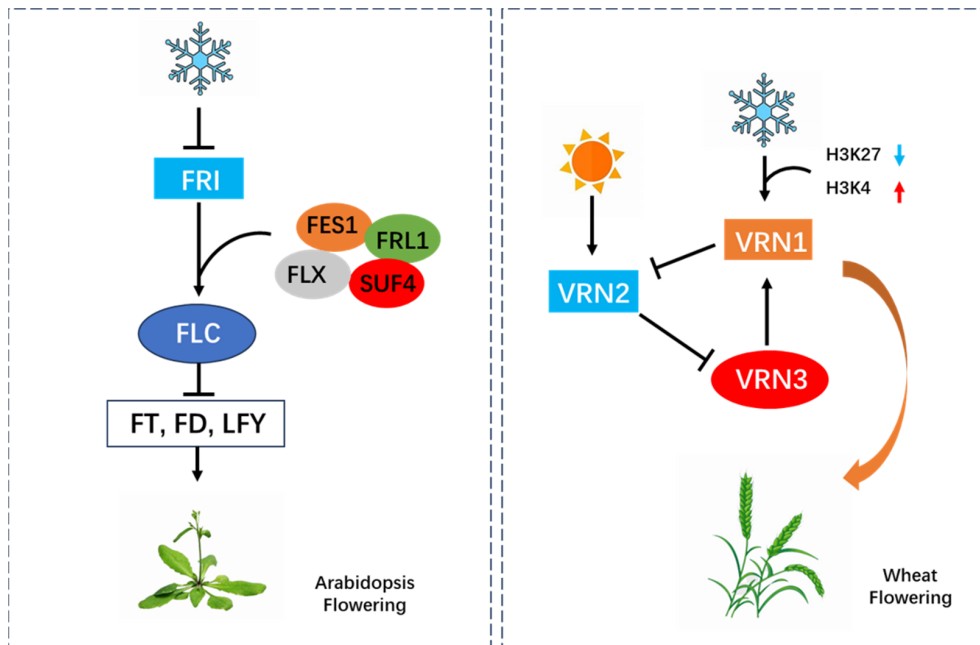

**Figure 2.** Vernalization pathways in *Arabidopsis* and wheat. Solid lines indicate identified connections, arrows indicate positive regulation, horizontal bars indicate negative regulation, red arrows indicate increases, and blue arrows indicate decreases. Left: *FRI* interacts with *FES1, FRL1, FLX,* and *SUF4* to activate the transcription of the downstream gene *FLC*, which regulates flowering in *Arabidopsis* by repressing the flowering integrative genes *FT, FD,* and *LFY*, and low temperature suppresses *FRI* expression [98,157,159,162–164]. Right: prior to vernalization, *VRN1* expression levels are low, and high levels of *VRN2* inhibit *VRN3* expression, thereby preventing flowering. During prolonged cold winter exposure, *VRN1* expression is activated by chromatin modification via decreased H3K27 methylation and increased H3K4 methylation, which downregulates *VRN2* expression and promotes the accumulation of *VRN3* in the leaf, and *VRN3* moves to the apical meristem to maintain high levels of *VRN1* and accelerate flowering [118,120,147,165–167].

Comparatively, genes associated with vernalization requirements vary from *Arabidopsis* to the winter cereals wheat and barley. Thus far, a model of vernalization has been established in wheat and barley in which *VERNALIZATION1 (VRN1)* (29, Table 1), *VRN2*, and *VRN3* are the core regulatory genes in the molecular framework of vernalization-responsive flowering time control (Figure 2). *VRN1* encodes a MADS-box transcription factor and acts as a plant flowering activator in grasses by vernalization-induced regulation of the plant transition from vegetative growth to reproductive development and is putatively presumed to be orthologous to the *Arabidopsis FRUITFULL (FUL)*, *AP1*, and *CAULIFLOWER (VAL)* (30–31, Table 1) genes. *VRN2*, a floral repressor encoding a CCT domain and zinc-finger containing protein, delays flowering until the plant's vernalization. *VRN3* is the orthologous gene of *Arabidopsis FT* encoding a polyethanolamine-binding protein whose expression is induced by the photoperiod and vernalization, and acts as a mobile florigen to accelerate the flowering of plants [168–171]. Prior to vernalization, which is the stage of vegetative growth in winter cereals, *VRN1* expression levels are low, whereas high levels of *VRN2* repress *VRN3* expression, thereby preventing flowering [172]. During prolonged winter cold exposure, *VRN1* expression is activated via chromatin modifications of reduced H3K27 methylation and increased H3K4 methylation, thereby downregulating the expression of *VRN2* and ultimately promoting *VRN3* accumulation in the leaves after vernalization [119,120,165]. Subsequently, *VRN3* in the leaves then moves to the shoot apical meristem in the form of a mobile florigen to maintain high levels of *VRN1* and accelerates flowering [165]. In recent years, the molecular mechanisms of the vernalization response in *Brachypodium distachyon*, which belongs to the same subfamily as wheat and barley, have been gradually revealed. The regulatory mechanisms of *VRN1* and *VRN3* are similar to those of wheat and barley, but *VRN2* can be induced under long-term low-temperature exposure, indicating that the molecular mechanisms of the vernalization response are not conserved despite being in the same subfamily [148]. The vernalization requirement is a crucial trait of the transition from the vegetative to the reproductive stage in many crop species, and, therefore, the studies of the molecular mechanisms underlying vernalization-regulated flowering in plants are of great significance for the understanding of low-temperature-induced flowering.

*3.2. Cold-Stress-Responsive Flowering*

The vernalization response in plants can only be achieved by exposure to winter cold for a sufficiently long period of time, whereas cold stress acclimation involves a rapid response to low, non-freezing temperatures. The vernalization pathway accomplishes developmental transitions mainly by accumulating prolonged cold exposure, whereas cold stress induces low-temperature tolerance in plants mainly by sensing low temperatures for a short period of time. Thus, unlike the extensive studies of the vernalization pathways in annual plants, the regulatory pathway of cold stress acclimation may be distinct. Cold acclimation is the cold or freeze stress tolerance acquired by plants exposed to low positive temperatures of 0–5 °C for short periods of time [173,174]. The adaptation of diverse temperature ranges is specific in various plant species, with 0–15 °C during rice growth considered to be cold or chilling stress, while near or below 0 °C is considered to be freezing stress. Additionally, *Arabidopsis* is adapted to growth in a different temperature range with rice, and the temperatures for cold (also referred to as chilling) and freezing stresses are 4–10 °C and below 0 °C, respectively [175–177].

During cold and freezing stress, plant growth and developmental processes are inhibited, but plants have evolved the competence to resist, tolerate, escape, or adapt to the stresses through biochemical, morphological, and transcriptional alterations to protect themselves from stressful environmental conditions. The biological membrane is the main damaged part of plants under low-temperature stress. Low temperature will reduce the fluidity of the membrane, thus enhancing the permeability of the membrane to electrolytes and other small molecules, resulting in an imbalance in ion exchange [66,178]. Also, low-temperature stress triggers lipid peroxidation. Malondialdehyde (MDA) is the final de-

composition product of membrane lipid peroxidation, which reflects the degree of damage to the plant and serves as one of the indicators of cold tolerance, and the accumulation of MDA will cause the disruption of the integrity of biological membranes, thereby altering the membrane permeability and affecting the normal physiological and biochemical reactions of plants [66,179,180]. Cold or chilling stress refers to the low, but non-freezing, temperature that occurs frequently in nature, which has a tremendous restrictive effect on the geographic location and crop yield of plants (especially tropical plants), leading to reduced membrane fluidity and cellular dysfunction, which can cause plant wilting, etiolation, and even necrosis, ultimately constraining growth and development [181–183]. Freezing stress refers to the subzero temperatures in nature that have a dramatic impact on plant photosynthesis, respiration, and metabolic processes, and mainly involves the formation of ice, which leads to cell dehydration and membrane damage, accompanied by alterations in calmodulin, intracellular $Ca^{2+}$ levels, ROS signaling, and phytohormones [173]. It should also be noted that the influence of low-temperature stress depends not only on the actual temperature, but also on whether the cold or freezing stress appears gradually or suddenly, and on the duration of the low-temperature stress. Additionally, the regulation of flowering time by the ambient temperature has been reported in multiple plants range from annuals to perennials, where short-term low-temperature stress can cause delayed flowering in a wide range of plants [23] (Table 2).

Plants regulate the activity of many classical floral pathway regulators, such as *AP1* [184], *FT* [46,164], *PHYTOCHROME-INTERACTING TRANSCRIPTION 4 (PIF4)* [123], *PIF5* [124], *MADSAFFECTING FLOWERING 2 (MAF2)* [109] (32–33, Table 1), and *SVP* [121], by sensing changes in the ambient temperature, thereby optimizing flowering time and improving cold acclimation or freezing tolerance (Figure 3). The exposure of vernalization-sensitive *Arabidopsis* to prolonged cold promotes floral transition via the vernalization pathway. By contrast, cold stress caused by short-term cooler temperatures delays flowering through activation of *FLC*. In *Arabidopsis*, *FLOWERING CONTROL LOCUS A (FCA)*, *FLOWERING LOCUS VE (FVE)*, *SVP*, and *FLOWERING LOCUS M (FLM)* (34–36, Table 1) are the cross-talk regulators between flowering time and the cold stress response. *SVP* prefers to bind to *FLM-β* to regulate low-temperature-responsive flowering [108]. *FCA* and *FVE* are floral-autonomous pathway genes and act to repress the expression of *FLC*, while *SVP* forms a flowering repressor complex with *FLC* to become a floral repressor, which ultimately co-regulates *FT* expression (Figure 3). The mutants *fca*, *fve*, and *svp* are insensitive to ambient low-temperature-induced flowering [23,185–187]. More recent studies have shown that the photoperiodic flowering pathway plays an important role in temperature perception, mediated by the circadian clock genes *GI* and *CO*, which, in turn, regulate the floral integrators *SOC1* and *FT*. The transcript levels of *GI* are elevated together with upregulation of cold-responsive genes under low-temperature stress (Figure 3). Compared with the control, *gi* mutants with a genetic background of the *Col-0* ecotype are more resistant to freezing stress [100], whereas *gi* mutants with a genetic background of the *Ler* ecotype have increased susceptibility to freezing stress and are defective in freezing tolerance [188]. Also, *GI* responds to temperature-regulated flowering time in *Medicago truncatula* [189]. Secondly, *CO* serves as a molecular link that combines cold signaling with the photoperiodic flowering pathway. The degradation of CO induced by low-temperature stress is mediated by the E3 ubiquitin ligase *HIGH EXPRESSION OF OSMOTICALLY RESPONSIVE GENE 1 (HOS1)* (37, Table 1), which, in turn, inhibits the *CO*-mediated activation of *FT* and ultimately regulates the delayed flowering of *Arabidopsis* [111] (Figure 3). Additionally, *HOS1* and *FVE* negatively regulate the cold stress response, with *fve* and *hos1* mutants having enhanced cold stress tolerance and *hos1* mutants exhibiting an early flowering phenotype. The chromatin modification factor *FVE* forms a histone repressor complex with *HDA6*, which inhibits *FLC* transcription by directly binding to *FLC* chromatin, ultimately leading to altered flowering time. Under short-term cold stress, *HOS1* induces *FLC* expression at the chromatin level by antagonizing the actions of *FVE* and *HDA6*, interfering with the association of *HDA6* with *FLC* chromatin in an *FVE*-dependent manner, which, in turn,

leads to late flowering [112,113]. Moreover, other transcription factors are also important for plants to regulate flowering in response to cold stress. *HOS15* can interact with *GI* and mediate GI degradation to repress flowering in response to low ambient temperature in *Arabidopsis* [114] (Figure 3). The *atho9* mutant exhibits a delayed flowering phenotype and is extremely sensitive to cold stress.

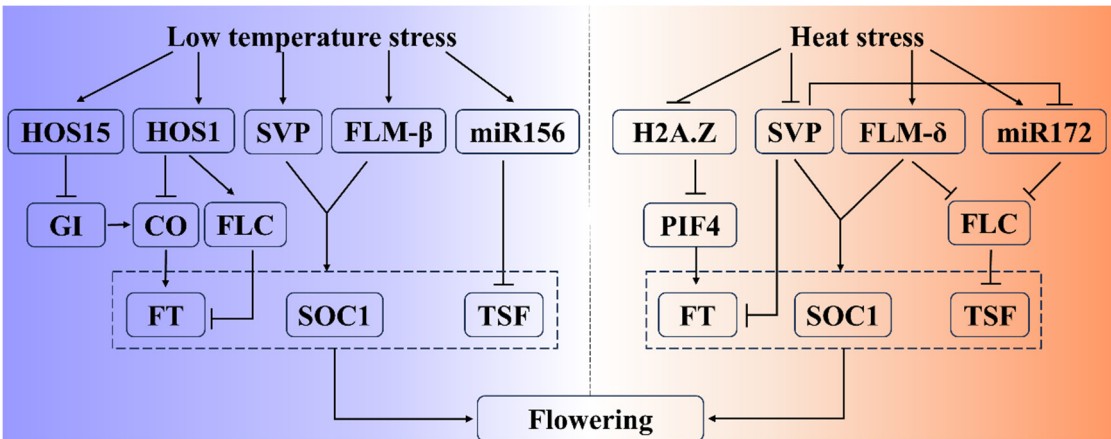

**Figure 3.** A simplified regulatory pathway for the link between ambient temperature and flowering in *Arabidopsis thaliana*. Low-temperature stress: *HOS15* and *HOS1* were able to degrade GI and CO proteins, respectively, indirectly inhibiting the expression of *FT* [111,114], in addition to *HOS1* positively regulating the expression of *FLC* [113], whereas *FLM-β* regulated the expression of *FT*, *SOC1*, and *TSF* by enhancing the stabilization of *SVP* [108,186]. miR156 directly targeted *TSF*, reducing its transcript abundance, and delayed flowering in *Arabidopsis* under low temperature [138]. Heat stress: under higher temperature conditions, miR172 negatively regulated the expression of *FLC* and deregulated the inhibitory effect of *FLC* on *TSF*. H2A.Z was repressed at high temperature, which attenuated its inhibitory effect on PIF4, resulting in an increase in the transcript level of *FT* [129]. *FLM-δ* accelerated the degradation of SVP at high temperatures, which deregulated the inhibitory effect on genes such as *FT*, *SOC1*, and *TSF* [108]. Solid lines indicate identified links, arrows indicate positive regulation, and horizontal bars indicate negative regulation.

In contrast, the short-day plant *Pharbitis nil*, when exposed to a temperature range of 16–18 °C shows restricted vegetative growth and is induced to flower; thus, the process is considered to be cold-stress-induced flowering [190]. *PnFT1* and *PnFT2*, two homologs of *Arabidopsis FT* in *Pharbitis* (Table 2), are major regulatory genes involved in low-temperature-induced flowering [30,31]. Further studies are required to elucidate the gene networks that regulate flowering under cold stress environmental conditions. The effect of cold stress on the floral transition has also been reported in perennials, such as chrysanthemum (*Chrysanthemum morifolium*) [110] and poplar (*Populus* spp.) trees [115] (Table 2). Similarly, among subtropical and tropical tree species, including macadamia, mango, avocado, lychee, lime, trifoliate orange, and satsuma mandarin, flowering is also induced by low-temperature stress [116,191–193]. The increase in flower density under cold stress is associated with upregulation of *CiFT* expression in citrus buds and leaves. The citrus transcription factor *CiNF-YA1* has been reported to control the response process to low-temperature stress by forming a complex with *CiNF-YB2* and *CiNF-YC2* as an upstream regulator of *CiFT* [92]. The CiFDα protein produced by *FD* alternative splicing is induced by low temperature, together with FT and the 14-3-3 proteins, forming a complex that regulates plant flowering [13]. Furthermore, two *FT* homologues in poplar trees have been shown to play important roles in coordinating the transition between vegetative growth and reproductive development, with *FT1* determining the initiation of reproductive development in response to low winter temperatures [115] (Table 2). However, the molecular mechanisms involved in the regulation of flowering by cold stress in perennials, especially in woody plants,

remain rudimentary compared to those in annual plants. Further studies are necessary to elucidate the specific gene networks that regulate flowering in perennial plants under cold-stress environmental conditions by referring to the relevant mechanisms involved in annual plants.

*3.3. Heat-Stress-Responsive Flowering*

With predicted global warming and climate change approaching, crop growth and development can be limited by high ambient temperature [194–196]. Heat stresses are considered to be above the normal optimal temperature and exceed the critical threshold of plant tolerance, which is ultimately sufficient to cause irreversible damage to plant growth and development [197]. The temperature threshold is defined as the optimal temperature range for regulating growth throughout the plant life cycle, which varies according to the plant species, developmental stage, and organ type. Warm-season crops, such as maize, soybean, tomato, and rice, generally exhibit higher temperature thresholds than barley and wheat from temperate zones. Many crops display slightly lower temperature thresholds at the early seed development and fruit-setting stages compared to other developmental phases [198,199]. Reproductive organs are more sensitive to moderate or extreme temperatures during plant growth and development compared to vegetative organs [200].

A recent review also indicates that how plants respond to high-temperature stress has become a hot topic of research [51]. Plants are sensitive to temperature, and heat stress affects a range of physiological, biochemical, morphological, and developmental processes, which, in turn, mitigate excessive damage from high temperature through such alterations [122,201]. Heat stress interferes with cellular homeostasis, mainly manifested as increased fluidity of lipid bilayers at high temperatures, increased membrane permeability, protein denaturation, organelle function destruction, and even cell death [202,203]. It has been reported that the nucleus and cytoplasm oxidation of *Arabidopsis thaliana* is increased under heat stress, as evidenced by elevated levels of $H_2O_2$, which, in turn, induces various reactive oxygen scavenging enzymes at the transcriptional level [178,204,205]. Moreover, heat stress can also induce accelerated plant growth, which, in turn, generates plant thermal adaptation by moving susceptible parts away from the stress, a process known as thermomorphogenesis [202,203,206–208]. For example, *Arabidopsis* display a tendency for hypocotyl and petiole elongation under moderately high ambient temperature conditions [201], whereas the stomata of barley, tomato, and *Arabidopsis* are induced to remain open under extreme heat conditions [194,209]. Similarly, monocotyledonous crops, such as rice, wheat, barley, and maize, or dicotyledonous crops, such as soybean and tomato, showed elongation of leaves and hypocotyls at moderately high temperatures, with tomato also exhibiting hypoplastic leaves [194,210–212]. Importantly, the series of changes that occur when plants are gradually or briefly exposed to high temperatures are conducive to the enhancement of heat stress tolerance, whereas the sudden occurrence and prolonged duration of heat stress can be detrimental to plant growth and development.

The reproductive development stage, especially the floral transition process, is greatly affected by high-temperature fluctuations compared to the adaptive growth of vegetative development under heat stress conditions [213–215]. Plants avoid high-temperature exposure during fertilization by altering flowering time under heat stress [216–218] (Table 2). Wheat, sorghum, and rice, for example, flower in the morning or in the cooler evening to complete fertilization before extreme temperatures lead to sterility [76,219,220]. Plant species that cannot change their flowering time in response to high-temperature fluctuations gradually disappear from their former natural habitats and eventually migrate to higher altitudes and latitudes [201]. Moreover, heat stress also has a tremendous effect on floral organ development in addition to flowering time. The adaptability of plants to high-temperature stress in the natural environment is different, which can be reflected in the great differences in flowering responses to heat stress. Heat stress has been reported to accelerate flowering in *Arabidopsis* and to delay flowering in *Brassica rapa* [129], suggesting that the effect of high ambient temperatures on flowering time is not a universal outcome.

High temperatures induce early flowering in the monocotyledonous bulb plant *Narcissus tazetta* [221,222], in contrast to elevated summer temperatures that delay flowering in *Chrysanthemum morifolium* [128,223]. The accelerated flowering of crops in response to high ambient temperature fluctuations does not translate into higher yields, due to the fact that accelerated flowering in response to heat stress is also accompanied by faster senescence, which leads to stunted grain growth, ultimately causing lower crop yields [224]. Therefore, understanding the specific mechanisms of plant flowering in response to heat stress is of great significance for improving adaptive growth at high temperatures and for more effectively balancing the relationship between plant tolerance to heat stress and normal growth at critical developmental stages.

The plant heat stress response (HSR) acquires high-temperature tolerance through the induction of heat shock proteins (HSPs) and heat stress transcription factors (HSFs) [225] (38–40, Table 1). Transcriptional regulation is extremely important in the plant response to heat stress, especially HSFs, which are responsible for the rapid transcriptional activation of downstream genes. The HSF-encoding genes are defined by 21 members in *Arabidopsis* and 25 members in rice, which are differentiated by the structural features of the oligomerized domains into A, B, and C categories [226]. Based on an analysis of HSFs in *Arabidopsis* and tomato, it was shown that the thermal-response-related genes of the HSF family are also closely related to plant reproductive development [227]. The development of gametophytes under heat stress in *Arabidopsis* is regulated by the homeostasis between *AtHSFB2a* and its natural antisense RNA [228]. In tomato (*Solanum lycopersicum*), moderate heat treatment (37.5 °C) leads to the accumulation of *SlHSFA2* and enhances the tolerance of seedlings under extreme heat stress conditions (47.5 °C). Meanwhile, *SlHSFA2* expression was inhibited when pollen was subjected to heat stress in the early developmental stage of pollen formation, thus reducing pollen viability and the germination rate, suggesting that *SlHSFA2* is an important regulator of heat tolerance during pollen development [229]. HSPs were initially recognized as proteins strongly induced by heat stress and which protected plants from stress by re-establishing normal protein conformation and cellular homeostasis under the control of HSFs [230]. The *Arabidopsis* BOBBER1 (AtBOB1) (41, Table 1) protein belongs to a class of small HSPs (sHSPs), and *atbob1* mutants have been reported to exhibit developmental defects in flowers and the inflorescence meristem and lower tolerance to high temperature, indicating that HSPs play a central role in both heat tolerance and the floral development of plants [231].

The molecular regulatory mechanisms of floral transition in response to heat stress differ in plants. The accelerated flowering in *Arabidopsis* under moderately high temperatures is due to increased *AtFT* expression [122,232]. *ft* mutants did not exhibit accelerated flowering in response to heat stress [233]. Flowering response to heat stress was also achieved in soybean by promoting *GmFT2a* and *GmFT5a* expression and repressing the upstream negative regulators *E1* and *E2* [125]. Similarly, high temperature, which delays flowering in chrysanthemum, is also associated with downregulation of *FLOWERING LOCUS T-like 3* (*FTL3*, an *FT* homologue) (42, Table 1) expression [128]. However, there is no difference in the expression levels of *FT1* in barley and wheat subjected to high-temperature treatment, indicating that the flowering of cereals responds to heat stress in an *FT*-independent manner [214] (Table 2). Knowledge about the specific mechanism of floral transition in *Arabidopsis* in a fluctuating high-temperature environment provides a framework for further research on flowering response to heat stress in different plants. At the molecular level, the histone variant *H2A.Z* decreases the incorporation of nucleosomes during flowering at higher temperatures, which, in turn, permits the binding of the *bHLH* transcription factor *PIF4* to the *FT* promoter, and ultimately accelerates flowering in a temperature-dependent manner. Neither *pif4* nor *pif5* mutants can accelerate flowering under heat stress, and *pif4 pif5* double mutants flower later compared with the single mutation, suggesting that *PIF4* and *PIF5* are related to high-temperature-responsive flowering [123,234]. Additionally, *PIF4* has been reported to be involved in controlling temperature-sensing memory and reproductive transition in *Arabidopsis* [123], and the *PIF4* homologue in soybean exhibits unique

high-temperature adaptability [235]. *SVP* tends to bind *FLM-δ* to regulate the expression of *FT*, *SOC1*, and *TSF*, ultimately mediating flowering in response to high temperature [108] (Figure 3).

Due to the influence of extreme high temperatures, the alteration of flowering time can have a huge impact on crop productivity. In barley, high-temperature-dependent *ELF3* increased transcription levels of the core circadian clock genes *GI*, *PSEUDO RESPONSE REGULATOR (PRR)*, and *LUX ARRHYTHMO (LUX)* [236,237] (43–44, Table 1). A late-flowering mutant allele, *Ppd-1*, was reported to retard flower development and reduce the flower and seed number in spring barley at high temperatures, in contrast to the *Ppd-1* and *elf3* mutants, both of which showed earlier flower development and unchanged seed number [237]. Moreover, an allelic variation of *ELF3* in wheat may underlie an Earliness *per se* locus (Eps-D1) (45, Table 1), which regulates flowering time depending on temperature [238,239]. High-temperature-delayed short-day flowering was also accompanied by increased expression of *HvODDSOC2*, the MADS box flowering suppressor associated with the *FLC* gene family [126]. Heat stress during flowering of rice, a short-day plant, can lead to sterility. *EXTRA GLUME 1 (EG1)* (46, Table 1), a gene encoding a lipase primarily located in mitochondria, acts upstream of flower identity genes (*OsG1*, *OsMADS1*, and *OsMADS6*) in a high-temperature-dependent manner, thereby promoting robust flower development [127]. In contrast, the *eg1* mutants exhibit high plasticity in spikelet development at high temperature and have a detrimental effect on maintaining flower development [127]. In maize, *ZmNF-YA3* binds to the *FT-like12* promoter to accelerate maize flowering, and the *Zmnf-ya3* mutant exhibits different high-temperature tolerance compared with the control, indicating that NF-Y transcription factors play an important role in maize flowering response to heat stress [106] (Table 2). In summary, studies on the mechanisms of flowering in response to heat stress vary widely between model plants and crops, and the regulatory networks that have been studied are still relatively basic. Therefore, future research on the floral transition caused by heat stress can enrich the overall regulatory network by cross-referencing the components of molecular mechanism studies in different plant species, laying an extraordinarily important foundation for research on perennial plants.

The effects of heat stress on the reproductive development of perennial plants, especially woody plants, are poorly understood. Woody plants growing in the temperate zone follow an annual growth cycle of sprouting in spring, flowering in summer, followed by the emergence of new buds in the fall [240]. However, the current global warming has a great impact on the timing of spring bud germination, so the control of temperature on the timing of bud burst in woody plants has become a new trend in research. In sexually mature poplar, *FT1* has been reported to initiate reproductive development in response to low temperatures, whereas *FT2* promotes vegetative growth as well as bud set inhibition under high-temperature conditions [115]. Additionally, *GtFT2* was found to be involved in dormancy regulation induced by temperatures in the perennial herb gentian, peaking at the time of dormancy release [241]. Due to the lack of available model plant species, the current understanding underlying the molecular regulation mechanisms of the bud dormancy cycle in woody plants is still incomplete. Similarly, due to the long study period and the fact that the effects of heat stress on reproductive growth in woody plants are not immediately apparent, the analysis of the mechanisms of this aspect in woody plants is limited. Taken together, future studies on the molecular mechanisms of flowering response to heat stress in woody plants can be further expanded by utilizing *Arabidopsis* as the framework. A detailed knowledge of the molecular mechanisms by which heat stress affects flowering is critical for breeding plant species with higher yields and greater tolerance to heat and provides crucial information for mitigating the effects of climate alteration.

## 4. Conclusions and Future Perspectives

The optimal flowering time is crucial for maximizing reproductive success, a process that is precisely regulated by a wide variety of environmental and internal factors. Accordingly, this review outlines the current knowledge showing that diverse abiotic stresses, such

as drought, low-temperature and high-temperature stress, alter the flowering time of plants. Meanwhile, the roles of the critical flowering regulation genes and stress-response-related genes are identified with consideration of the cross-talk molecular mechanisms underlying flowering time regulation and the stress response. Compared with annual plants, especially the model plant *Arabidopsis*, perennials, especially woody plants, involve a longer period of time, and a heavier workload, to study the molecular mechanisms related to reproductive development, resulting in the fact that little is known about abiotic stresses controlling flowering time in perennial plants. Hence, it is important to understand the molecular mechanisms of flowering time regulation in perennials under environmental stress. Although our understanding underlying the effects of abiotic stress on flowering timing has increased considerably in perennials, most of the regulatory mechanisms associated with abiotic stress have been identified in *Arabidopsis* and currently remain elusive in perennials. More importantly, the molecular pathways by which abiotic stress affects flowering time in different plants are similar, but also exhibit many unique aspects. In future studies, it will be both challenging and necessary to utilize the molecular network obtained in *Arabidopsis* to advance the understanding of the molecular pathways involved in perennial plants flowering time in response to abiotic stresses, as well as to conduct comparative studies on different plant species.

Flowering time is affected by abiotic stresses through a complex network, but most of the existing studies focus on the role of individual genes in the stress-regulated flowering process, which makes it difficult to localize the key factors and is insufficient to show that the whole process is controlled by one pathway only. Currently, multi-omics combined analysis can provide in-depth analysis of complex regulatory networks, including gene expression, protein function, and metabolites [242–244]. Comparative transcriptomics has revealed differentially expressed genes for drought-stress-induced flower formation in *Curcuma kwangsiensis* [74], as well as a regulatory cascade for flower development under drought stress in *Arabidopsis thaliana* [40]. Meanwhile, the detection of protein changes under abiotic stresses by comparative proteomics analysis is essential for revealing stress-induced signal perception as well as response, and this information cannot be revealed by transcriptomic analysis [245]. Certainly, metabolomics is also an important molecular tool for identifying plant responses to different abiotic stresses [246,247]. However, the application of omics approaches to abiotic-stress-regulated flowering time in plants is relatively rare, so the combination of different omics tools is a new trend for future research to identify complex regulatory networks involved in the abiotic stress regulation of plant flowering. Since environmental alteration generally entails the simultaneous existence of multiple abiotic stress factors, it is also necessary to focus on the effects of the combination of various abiotic stresses on plant flowering in the future, so as to provide better tools for breeders to cultivate reproductive traits in response to environmental alterations.

**Author Contributions:** M.C. and T.-L.Z. performed different aspects of the research; J.-Z.Z. and C.-G.H. designed the research; M.C. wrote the manuscript. All authors have read and agreed to the published version of the manuscript.

**Funding:** This research was supported financially by the National Natural Science Foundation of China (grant nos. 319702356, 32072521, and 31872045).

**Conflicts of Interest:** The authors have no conflict of interest to declare.

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
