# Peer review of "The Role of Drought and Temperature Stress in the Regulation of Flowering Time in Annuals and Perennials"

_agronomy, doi:10.3390/agronomy13123034_

Round 1

Reviewer 1 Report

Comments and Suggestions for Authors

Regarding this manuscript, I do not recommend its acceptance, considering the comments as follows:

  1. There are other reviews already published on this research topic. By the way, these review articles were not cited by the authors.

For example:

https://doi.org/10.3389/fpls.2022.1052660    “Plant responses to high temperature and drought: A bibliometrics analysis”

Therefore, I am not sure that this manuscript meets the novelty criterion.

  1. There are several topics that are not addressed by the authors (such as important biochemical, molecular and physiological insights - for example, related to omic approaches, antioxidants, etc.).

  2. The authors should mention the selection criterion of the articles, considering that I did not find many articles on this research topic. Indeed, the authors should cite as many articles as possible on this topic.

  3. I did not understand the selection criterion of the articles to be cited in the table provided. There are many other articles that could be cited as well as columns that could be incorporated to make the table more informative.

4.1) The table is very short. Many more articles should be cited.

  1. The authors should incorporate more figures to make the article more illustrative and more tables to make the manuscript more informative to the readers.

  2. There are many recent articles (with the year of publication in 2023 and 2024) that were not cited.

  3. The manuscript should be better divided into more subtopics.

  4. In many parts the manuscript seems to be merely informative like a book chapter instead of a review article with a good critical argumentation and new perspective.

  5. The title needs to be improved and more informative.

Other comments include:

The authors could incorporate a list of abbreviations.

The authors could incorporate a high quality figure (like a graphical abstract) summarizing the review.

Author Response

Dear reviewer,

Thank you very much for your careful reading, helpful comments, and constructive suggestions, which has significantly improved the presentation of our manuscript (agronomy-2722178). We have carefully considered all comments from the reviewers and revised manuscript accordingly. Please find the detailed responses below and the corresponding revisions in the re-submitted files.

Comments 1: There are other reviews already published on this research topic. By the way, these review articles were not cited by the authors. For example: https://doi.org/10.3389/fpls.2022.1052660 “Plant responses to high temperature and drought: A bibliometrics analysis” Therefore, I am not sure that this manuscript meets the novelty criterion.

Response 1: Thank you for pointing this out. We are very sorry for our negligence of reference citation. We have revised the manuscript accordingly. (Line 155-156, Line 635-636)

And, we also cite as many articles as possible on this research topic, including many recent articles (with publication years of 2022 and 2023).

Regarding novelty, we focus on the effects of abiotic stress (mainly drought, low temperature and high temperature) on flowering time of plants, that is, the physiological and biochemical responses and molecular regulatory mechanisms of early or late flowering of annual plants or perennial plants under abiotic stress conditions. This differs from the review, https://doi.org/10.3389/fpls.2022.1052660 “Plant responses to high temperature and drought: A bibliometrics analysis”, in that the regulation of drought and temperature on flowering time of plants and their molecular mechanisms are discussed. We also cite some recent review articles on this topic (Chirivì et al., Molecular links between flowering and abiotic stress response: a focus on Poaceae. Plants 2023,12,331, doi:10.3390/plants12020331; Takeno, K. Stress-induced flowering: the third category of flowering response. J. Exp. Bot. 2016,67,4925-4934, doi:10.1093/jxb/erw272; Ma et al., Molecular genetic analyses of abiotic stress responses during plant reproductive development. J. Exp. Bot. 2020,71,2870-2885, doi:10.1093/jxb/eraa089; Jacott et al., Feeling the heat: developmental and molecular responses of wheat and barley to high ambient temperatures. J. Exp. Bot. 2020,71,5740-5751, doi:10.1093/jxb/eraa326.), but unlike them, we further summarize the studies on the effects of drought and temperature on flowering time in perennial plants, especially woody plants. Considering the time-consuming and low yield of studies on the environmental effects on the reproductive development of woody plants, it is hoped that the content of this review can provide useful research ideas for perennial plants after summarizing related studies on annual and perennial plants.

To complement the content on plant response to abiotic stresses, we have cited additional relevant articles. For example, Duan et al., Testing the limits of plant drought stress and subsequent recovery in four provenances of a widely distributed subtropical tree species. Plant Cell Environ. 2022,45,1187-1203, doi:10.1111/pce.14254. Shao et al., Physiological and biochemical dynamics of Pinus massoniana Lamb. seedlings under extreme drought stress and during recovery. Forests 2022,13,65, doi:10.3390/f13010065.

We thank the reviewer for this input and realize that the novelty may not have been stated sufficiently clear in the manuscript before. We have improved the manuscript, especially in the title (Line 2-3), abstract (Line 9-28) and conclusion (Line 773-815) in order to make the contents of the manuscript more specific.

Comments 2: There are several topics that are not addressed by the authors (such as important biochemical, molecular and physiological insights - for example, related to omic approaches, antioxidants, etc.).

Response 2: We sincerely appreciate the valuable comments. We have revised the manuscript in the corresponding positions.

Line 180-193: “Biochemically, it is manifested by the high expression of some drought resistance genes that positively increase the content of amino acids and sugars in plants (such as proline and trehalose), the enhanced activity of antioxidant-related enzymes, and the inhibition of the activity of enzymes involved in degradation pathways to ensure that normal metabolic homeostasis is maintained under drought stress condition [63]. Reactive oxygen species (ROS), including superoxide radicals (O2-), hydrogen peroxide (H2O2), and hydroxyl radicals (OH), regulate plant growth and development at lower concentrations [64,65]. Excessive accumulation of ROS under drought stress leads to membrane lipid peroxidation [66,67]. Previous studies have shown that excessive ROS in plants will be scavenged by antioxidant mechanisms, including enzymatic antioxidants SOD (superoxide dismutase), CAT (catalase), POD (peroxidase) and non-enzymatic antioxidants ascorbic acid, proline, flavonoids and polyphenols, which ultimately improve the plant drought resistance [68-71]. Ascorbic acid has also been reported to play a role in controlling flowering time in plants [72].”

Line 517-529: “During cold and freezing stress, plant growth and developmental processes are inhibited, but they have evolved the competence to resist, tolerate, escape, or adapt to the stresses through biochemical, morphological, and transcriptional alterations to protect themselves from stressful environmental conditions. Biological membrane is the main damaged part of plants under low temperature stress. Low temperature will reduce the fluidity of the membrane, thus enhancing the permeability of the membrane to electrolytes and other small molecules, resulting in the imbalance of ion exchange [66,160]. Also, low temperature stress triggers lipid peroxidation. Malondialdehyde (MDA) is the final decomposition product of membrane lipid peroxidation, which reflects the degree of damage to the plant and serves as one of the indicators of cold tolerance, and the accumulation of MDA will cause the disruption of the integrity of biological membranes, thereby altering the membrane permeability and affecting the normal physiological and biochemical reactions of plants [66,161,162].”

Line 636-644: “Plants are sensitive to temperature, and heat stress affects a range of physiological, biochemical, morphological, and developmental processes, which in turn mitigate excessive damage from high temperature through such alterations [196,197]. Heat stress interferes with cellular homeostasis, mainly manifested as increased fluidity of lipid bilayers at high temperatures, increased membrane permeability, protein denaturation, organelle function destruction and even cell death [198,199]. It has been reported that the nucleus and cytoplasm oxidation of Arabidopsis thaliana is increased under heat stress, as evidenced by elevated levels of H2O2, which in turn induces various reactive oxygen scavenging enzymes at the transcriptional level [160,200,201].”

Line 796-811: “Flowering time is affected by abiotic stresses as a result of a complex network, but most of the existing studies focus on the role of individual genes in the stress-regulated flowering process, which makes it difficult to localize the key factors and insufficient to show that the whole process is controlled by one pathway only. Currently, multi-omics combined analysis can provide in-depth analysis of complex regulatory network from gene expression, protein function and metabolites [242-244]. Comparative transcriptomics revealed differentially expressed genes for drought stress-induced flower formation in Curcuma kwangsiensis [74], as well as a regulatory cascade for flower development under drought stress in Arabidopsis thaliana [40]. Meanwhile, detection of protein changes under abiotic stresses by comparative proteomics analysis is essential for revealing stress-induced signal perception as well as response, and this information cannot be revealed by transcriptomic analysis [245]. Certainly, metabolomics is also an important molecular tool for identifying plant responses to different abiotic stresses [246,247]. However, the application of omics approaches to abiotic stress-regulated flowering time in plants is relatively rare, so the combination of different omics tools is a new trend for future research to identify complex regulatory networks involved in abiotic stress regulation of plant flowering.”

Comments 3: The authors should mention the selection criterion of the articles, considering that I did not find many articles on this research topic. Indeed, the authors should cite as many articles as possible on this topic.

Response 3: We sincerely appreciate the valuable comments. We have cited as many articles on this research topic as possible in the resubmitted manuscript.

The selection criteria for article citations have been explained in the resubmitted manuscript. (Line 101-105)

Comments 4: I did not understand the selection criterion of the articles to be cited in the table provided. There are many other articles that could be cited as well as columns that could be incorporated to make the table more informative.

4.1) The table is very short. Many more articles should be cited.

Response 4: We sincerely appreciate the valuable comments. We have described the selection criteria for literature citation in detail (Line 101-105) and have reworked the table to make them more informative. (Table 2, Line 324)

Comments 5: The authors should incorporate more figures to make the article more illustrative and more tables to make the manuscript more informative to the readers.

Response 5: We sincerely appreciate the valuable comments. We have added Figure 2 (Line 459) and Table 1 (Line 114) to make the article more illustrative and more informative to the readers.

For a list of abbreviations, see Table 1. (Line 114)

The effects of drought, low temperature and high temperature stress on flowering time of different plants and the genes related to flowering pathways involved are listed in Table 2. (Line 324)

Simplified regulatory pathways linking drought stress and flowering in Arabidopsis are shown in Figure 1. (Line 272)

The Arabidopsis and wheat vernalization pathways are shown in Figure 2. (Line 459)

Simplified regulatory pathways for the link between ambient temperature and flowering in Arabidopsis thaliana are shown in Figure 3. (Line 584)

Comments 6: There are many recent articles (with the year of publication in 2023 and 2024) that were not cited.

Response 6: Thank you for pointing this out. We are very sorry for our negligence of reference citation. We have cited more recent articles (with publication years of 2022 and 2023) in the resubmitted manuscript.

Comments 7: The manuscript should be better divided into more subtopics.

Response 7: Thank you for pointing this out. We have renumbered the sections in the corresponding positions. (Line 131, 197, 261, 406, 421, 502, 622, 773)

Comments 8: In many parts the manuscript seems to be merely informative like a book chapter instead of a review article with a good critical argumentation and new perspective.

Response 8: We sincerely appreciate the valuable comments. We have revised the manuscript in the corresponding positions.

Line 220-222: “The role of drought in influencing flowering time varies among plant species and environmental conditions, so that drought regulation of flowering is the result of multiple factors.”

Line 231-233: “Artificial control of the duration and intensity of drought treatment in agricultural production exhibits an important role in accelerating plant development, especially in the area of flowering.”

Line 254-260: “These indicate that the effect of drought stress on flowering time is not specific to a particular plant species, but is conserved in annuals as well as perennials. There are few studies on the regulation of flowering time by drought stress in perennials due to the long study time and lack of phenotype, but the available evidence can support the feasibility of this research topic in perennials. Moreover, studies on drought-induced flowering in perennial plants can be carried out on the basis of sufficient theoretical evidence in annual plants.”

Line 333-338: “It is evident that NF-YAs are likely to be functionally conserved in regulating flowering in annual and perennial plants, with functional diversity resulting from physiological differences in response to stress. These studies support a critical role for NF-YAs in promoting not only flowering time but also drought response (tolerance/sensitivity). However, future studies are needed to clarify whether NF-YAs are directly involved in regulating drought-affected flowering.”

Line 352-358: “Therefore, drought-induced flowering regulation genes is conserved between annual and perennial plants. Together, these studies strongly support the pivotal roles of flowering time-regulated genes in drought stress response and tolerance. The main challenge in woody plants is that the regulatory role of genes in drought-induced flowering can be demonstrated, but there is no phenotypic evidence of flowering, which is related to its own longer developmental process.”

Line 785-786: “Hence, it is important to understand the molecular mechanisms of flowering time regulation in perennials under environmental stress.”

Line 789-791: “More importantly, the molecular pathways by which abiotic stress affects flowering time in different plants are similar, but also exhibit many unique aspects.”

Line 796-799: “Flowering time is affected by abiotic stresses as a result of a complex network, but most of the existing studies focus on the role of individual genes in the stress-regulated flowering process, which makes it difficult to localize the key factors and insufficient to show that the whole process is controlled by one pathway only.”

Line 808-811: “However, the application of omics approaches to abiotic stress-regulated flowering time in plants is relatively rare, so the combination of different omics tools is a new trend for future research to identify complex regulatory networks involved in abiotic stress regulation of plant flowering.”

Comments 9: The title needs to be improved and more informative.

Response 9: We sincerely appreciate the valuable comments. We have revised the title in the corresponding positions.

Line 2-3: “The role of drought and temperature stress in the regulation of flowering time in annuals and perennials.”

Other comments include:

The authors could incorporate a list of abbreviations.

Response: Thank you for pointing this out. We have added Table 1 for a list of abbreviations. (Line 114)

The authors could incorporate a high quality figure (like a graphical abstract) summarizing the review.

Response: We sincerely appreciate the valuable comments. As for the manuscript, we summarized the relevant content through two tables (Line 114, 324) and three figures (Line 272, 459, 584). Considering the journal's typographical and formatting requirements, it is possible that the graphical abstract may not meet the expectations.

Reviewer 2 Report

Comments and Suggestions for Authors

Review comments

1. Line 2: The title of the article should be more descriptive. 2. Line 18-20: At the end of the abstract section, a few more sentences about the importance of this review and its contribution to the literature can be mentioned. 3. In the text, reference numbers should be placed in square brackets [ ], and placed before the punctuation (See journal instructions). 4. Line 101-103, 134-136, 251-253, 267-272, 341-343, 392-395, 407-411, 724-728Need citation. Adding the citations at the end of the sentences would be more appropriate. 5. Line 34, 96, 161, 228, 242, 324, 416, 422, 433, 434, 458, 486Some old references need to be changed to current ones. 6. Line 263: Add a citation for Figure 1 if available. 7. Line 18-20: “Better understanding of the role of crucial flowering regulatory genes in response to abiotic stresses from the crosstalk molecular mechanisms of flowering time regulation and stress response.” This sentence can be revised and detailed. 8. Line 203, 278, 557: “Arabidopsis” should be written italicized in the text.

There are some mistakes in the references section:

9. Line 788813, 816, 820, 824, 827, 834, 840, 844, 849, 862, 863, 867…: Latin names (Arabidopsis thaliana, Oryza sativa indica, Oryza glaberrima, Brassica rapa, Brassica napus, Hordeum vulgare, Zea mays etc.) should be written italicized. 10. All references should be revised according to the instructions of Agronomy Journal and references should be numbered.

Author Response

Dear reviewer,

Thank you very much for your careful reading, helpful comments, and constructive suggestions, which has significantly improved the presentation of our manuscript (agronomy-2722178). We have carefully considered all comments from the reviewers and revised manuscript accordingly. Please find the detailed responses below and the corresponding revisions in the re-submitted files.

Comments 1: Line 2: The title of the article should be more descriptive.

Response 1: We sincerely appreciate the valuable comments. We have revised the title in the corresponding positions.

Line 2-3: “The role of drought and temperature stress in the regulation of flowering time in annuals and perennials.”

Comments 2: Line 18-20: At the end of the abstract section, a few more sentences about the importance of this review and its contribution to the literature can be mentioned.

Response 2: We sincerely appreciate the valuable comments. We have revised the manuscript in the corresponding positions.

Line 25-27: “This review aims to clarify the effects of abiotic stresses (mainly drought and temperature) on plant flowering, which is significant for future productivity increases under unfavorable environmental conditions.”

Comments 3: In the text, reference numbers should be placed in square brackets [ ], and placed before the punctuation (See journal instructions).

Response 3: We gratefully appreciate for your valuable comments. We have revised reference numbers according to the reviewer’s comments in the corresponding positions. Please find the corresponding revisions in the re-submitted files.

Comments 4: Line 101-103, 134-136, 251-253, 267-272, 341-343, 392-395, 407-411, 724-728: Need citation. Adding the citations at the end of the sentences would be more appropriate.

Response 4: Thank you for pointing this out. We are very sorry for our negligence of references citation. We have revised the manuscript in the corresponding positions. (Line 45-47, Line 98-101, Line 137-139, Line 262-265, Line 359-361, Line 409-411, Line 424-427, Line 623-627, Line 636-641, Line 681-682, Line 687-689)

Comments 5: Line 34, 96, 161, 228, 242, 324, 416, 422, 433, 434, 458, 486: Some old references need to be changed to current ones.

Response 5: Thank you for pointing this out. We have changed the old references to current ones in the corresponding positions.

Comments 6: Line 263: Add a citation for Figure 1 if available.

Response 6: Thank you for pointing this out. We have revised the manuscript in the corresponding positions.

Line 273-282: “Figure 1. Simplified regulatory pathways linking drought stress and flowering in Arabidopsis thaliana. Drought escape: ABF3/ABF4 further activates the expression of LFY, AP1 and SOC1 by targeting NF-YC [95]. GI accelerates flowering under drought conditions by positively regulating the expression of CO and miR172 [94], which in turn activates the expression of FT, or by directly activating the transcription of TSF, which ultimately up-regulates the expression levels of LFY, AP1 and SOC1 [11,32,46]. Drought tolerance: miR169 targets NF-YA2 to reduce its transcriptional abundance [96], which attenuates the repressive effect on downstream genes FLC and SVP [97], while FRI positively regulates the expression of FLC and SVP, resulting in the repression of FT transcription and delayed flowering under drought conditions [98]. Solid lines indicate identified associations, arrows indicate positive regulation, and horizontal bars indicate negative regulation.”

Also, we have added a citation for Figure 2 and Figure 3.

Line 460-471: “Figure 2. Vernalization pathways in Arabidopsis and wheat. Solid lines indicate identified connections, arrows indicate positive regulation; horizontal bars indicate negative regulation, red arrows indicate increases, and blue arrows indicate decreases. Left: FRI interacts with FES1, FRL1, FLX and SUF4 to activate the transcription of the downstream gene FLC, which regulates flowering in Arabidopsis by repressing the flowering integrative genes FT, FD, and LFY, and low temperature suppresses FRI expression [98,136,138,141-143]. Right: prior to vernalization, VRN1 expression levels are low, and high levels of VRN2 inhibit VRN3 expression, thereby preventing flowering. During prolonged cold winter exposure, VRN1 expression is activated by chromatin modification via decreased H3K27 methylation and increased H3K4 methylation, which down-regulates VRN2 expression and pro-motes the accumulation of VRN3 in the leaf, and VRN3 moves to the apical meristem to maintain high levels of VRN1 and accelerate flowering [126,144-148].”

Line 585-596: “Figure 3. A simplified regulatory pathway for the link between ambient temperature and flowering in Arabidopsis thaliana. Low-temperature stress: HOS15 and HOS1 were able to degrade GI and CO proteins, respectively, indirectly inhibiting the expression of FT [177,180], in addition to HOS1 positively regulating the expression of FLC [179], whereas FLM-β regulated the expression of FT, SOC1, and TSF by enhancing the stabilization of SVP [171,173]. miR156 directly targeted TSF, reducing its transcript abundance, and delayed flowering in Arabidopsis under low temperature [115]. Heat stress: under higher temperature conditions, miR172 negatively regulated the expression of FLC and deregulated the inhibitory effect of FLC on TSF. H2A.Z was repressed at high temperature, which attenuated its inhibitory effect on PIF4, resulting in an increase in the transcript level of FT [181]. FLM-δ accelerated the degradation of SVP at high temperatures, which deregulated the inhibitory effect on genes such as FT, SOC1 and TSF [171]. Solid lines indicate identified links, arrows indicate positive regulation, and horizontal bars indicate negative regulation.”

Comments 7: Line 18-20: “Better understanding of the role of crucial flowering regulatory genes in response to abiotic stresses from the crosstalk molecular mechanisms of flowering time regulation and stress response.” This sentence can be revised and detailed.

Response 7: We sincerely appreciate the valuable comments. We have revised the manuscript in the corresponding positions.

Line 19-22: “Starting from the perspective of functional analysis of key flowering-regulated genes, it is of great help for researchers to quickly gain a deeper understanding of the regulatory effects of abiotic stress on the flowering process, to elucidate the molecular mechanisms, and to improve the regulatory network of abiotic stress-induced flowering.”

Comments 8: Line 203, 278, 557: “Arabidopsis” should be written italicized in the text.

Response 8: We sincerely appreciate the valuable comments. We have made corrections according to the reviewer’s comments in the corresponding positions.

Comments 9: Line 788, 813, 816, 820, 824, 827, 834, 840, 844, 849, 862, 863, 867…: Latin names (Arabidopsis thaliana, Oryza sativa indica, Oryza glaberrima, Brassica rapa, Brassica napus, Hordeum vulgare, Zea mays etc.) should be written italicized.

Response 9: We sincerely appreciate the valuable comments. We have made corrections at the appropriate positions in the manuscript. Also, the gene names in the manuscript as well as in the references were revised to italics.

Comments 10: All references should be revised according to the instructions of Agronomy Journal and references should be numbered.

Response 10: We sincerely appreciate the valuable comments. We have revised all references according to the instructions of Agronomy Journal in the corresponding positions of the manuscript.

Reviewer 3 Report

Comments and Suggestions for Authors

Dear authors,

thank you for your comprehensive review of flowering and impacts of drought and temperature stresses. I think you've done a good job of outlining the importance of the work you summarize and the need for continued research to investigate the details. I also really like how you've started each major topic (drought, temperature) with the more general information and then worked your way towards the detailed molecular findings. I have described some areas of improvement below, and I think that if these are addressed, then I can recommend this paper for publication.

What I feel is lacking in some sections is the actual effect of drought stress on flowering. In some cases it's clear that a specific gene regulates flowering, is also regulated by stress, and then what the effect of the stress is on flowering (i.e. how the flowering pathway is affected by the stress and how this then changes the timing of flowering). In other areas, it's not clear.  Generally, a lot of the manuscript seems to just point to the overlap in flowering and drought/temperature pathways, but doesn't clearly indicate the final consequence of the stress on flowering. One of the purposes of the study was to indicate how the plants balance the stress response with flowering, but I don't feel this is clearly and consistently described. Lines 351-354, for example, are clear and the purpose is conveyed. Lines 281-286, however, feel incomplete. It's clear in these lines that there is overlap between drought-responsive and flowering genes, but the full picture of how flowering is affected is not clear. Generally, the sections on temperature are much more clear.  If, in some cases, the purpose is solely to point out the overlap in genes between stress and flowering pathways because there isn't any literature to show how flowering is affected, then this should be pointed out and made clear.

I would suggest renumbering the sections. I think it would more clearly show the relationship between the different sections if they were numbered as 2a, 2b, 2c, etc... through the drought sections, and then 3a, 3b,... in the temperature sections, with the conclusions being section '4'. Also, for each topic of drought and temperature, there's an introductory paragraph that feels too small for its own number. For example, sections 2 and 6 as you currently have them. These could be renumbered to '2' and '3', with the sections following being numbered as 2a (instead of 3), 2b (instead of 4), etc... and then 3a (instead of 7), 3b (instead of 8), etc...  Perhaps this is outside of the control of the authors and is required by the journal, but if it's possible to change the sections I think it would flow a bit better.

I have not provided detailed editorial revisions since I believe the journal staff will take care of it, but in cases where the wording affects the science I have given some minor revisions below:

line 197 - instead of 'biofuels as well as forage crops', revise to 'forage and biofuel crops'

line 230 - remove 'thus' since this sentence is not a continuation of any former information, but is the start of a new paragraph. 

line 231 - remove 'however'; the second sentence agrees with the first, so there is no need for 'however'.

The section starting at line 267 is a bit confusing. The information is there it seems, but is a bit disjointed, and in some cases it feels like the end of the thought is missing. Often, a flowering gene is referred to and then the effect of stress on the expression of the gene is described, but then there's no information about how flowering is affected. As a reader, I kept asking myself, and what about flowering? If the expression is affected by the stress, and it has X effect on the stress response, then what's the effect on flowering? Line 317 is a great example of what there could be more of - it's clear that the gene functions via drought to accelerate flowering. I would take this as an example of how things could be made clear in other sections. And if the answer is that you don't know what the effect of the stress is on flowering, then that point should be made clear. 

line 279-281 - figure 1 is referenced here but WRKY44 is not in the figure. It should be made clear where exactly WRKY44 fits into the figure (if you can indicate a gene that it regulates, or is regulated by, in the figure then that would be sufficient.

line 283 - again, the connection isn't quite clear here. You mention SVP and then mention the ABA pathway, but I think this should be elaborated upon because it isn't clear what the final consequence is. What happens, with respect to flowering, when the ABA pathway genes are inhibited? It just needs a small bit of extra information to complete the thought. 

Line 285 - similar to above - you mention that the loss of FLC but the sentence, or idea, feels unfinished. loss of FLC leads to decreased drought tolerance, and what happens with flowering? 

Lines 749-750 - remove 'salt stress as well as well as poor nutrition stress' - this manuscript did not focus on those, so it doesn't seem like any conclusions should be made regarding them.

Comments on the Quality of English Language

Overall, the quality of English is fairly good, but I do suggest that the manuscript be checked over for small grammatical and structural revisions.

Author Response

Dear reviewer,

Thank you very much for your careful reading, helpful comments, and constructive suggestions, which has significantly improved the presentation of our manuscript (agronomy-2722178). We have carefully considered all comments from the reviewers and revised manuscript accordingly. Please find the detailed responses below and the corresponding revisions in the re-submitted files.

Comments 1: Thank you for your comprehensive review of flowering and impacts of drought and temperature stresses. I think you've done a good job of outlining the importance of the work you summarize and the need for continued research to investigate the details. I also really like how you've started each major topic (drought, temperature) with the more general information and then worked your way towards the detailed molecular findings. I have described some areas of improvement below, and I think that if these are addressed, then I can recommend this paper for publication.

Response 1: Thank you again for your professional review work on our manuscript and giving the above positive comments. As you are concerned, there are several problems that need to be addressed. According to your nice suggestions, we have made extensive corrections to our previous manuscript, the detailed corrections are listed below.

Comments 2: What I feel is lacking in some sections is the actual effect of drought stress on flowering. In some cases it's clear that a specific gene regulates flowering, is also regulated by stress, and then what the effect of the stress is on flowering (i.e. how the flowering pathway is affected by the stress and how this then changes the timing of flowering). In other areas, it's not clear. Generally, a lot of the manuscript seems to just point to the overlap in flowering and drought/temperature pathways, but doesn't clearly indicate the final consequence of the stress on flowering. One of the purposes of the study was to indicate how the plants balance the stress response with flowering, but I don't feel this is clearly and consistently described. Lines 351-354, for example, are clear and the purpose is conveyed. Lines 281-286, however, feel incomplete. It's clear in these lines that there is overlap between drought-responsive and flowering genes, but the full picture of how flowering is affected is not clear. Generally, the sections on temperature are much more clear. If, in some cases, the purpose is solely to point out the overlap in genes between stress and flowering pathways because there isn't any literature to show how flowering is affected, then this should be pointed out and made clear.

Response 2: We sincerely appreciate the valuable comments. We have revised the manuscript in the corresponding positions.

Line 293-295: “It has been confirmed that GI-miR172 pathway was involved in drought-induced early flowering escape by down-regulating WRKY44.”

Line 295-299: “Several other flowering inhibition genes are also induced by drought stress. For example, water deficit induces the flowering repressor gene SVP, which represses the transcription of genes related to ABA catabolism, and increases ABA accumulation, which improves drought tolerance in Arabidopsis, but flowering is delayed.”

Line 299-301: “Similarly, FLC, a flowering suppressor gene, also plays a role in drought stress pathway, and the loss of FLC function leads to early flowering and decreased drought tolerance in Arabidopsis.”

We have deleted lines 301-307 because that portion of the text does not fit the logic of the subject matter.

In conjunction with the theme of this section, we have added the following for logical soundness. Line 318-321: “These results suggest that when plants are subjected to drought stress, a large number of genes are induced to be expressed, including genes critical to the flowering pathway, and that differences in the expression of these genes between species ultimately lead to different flowering outcomes.”

Comments 3: I would suggest renumbering the sections. I think it would more clearly show the relationship between the different sections if they were numbered as 2a, 2b, 2c, etc... through the drought sections, and then 3a, 3b,... in the temperature sections, with the conclusions being section '4'. Also, for each topic of drought and temperature, there's an introductory paragraph that feels too small for its own number. For example, sections 2 and 6 as you currently have them. These could be renumbered to '2' and '3', with the sections following being numbered as 2a (instead of 3), 2b (instead of 4), etc... and then 3a (instead of 7), 3b (instead of 8), etc...  Perhaps this is outside of the control of the authors and is required by the journal, but if it's possible to change the sections I think it would flow a bit better.

Response 3: Thank you for pointing this out. We have renumbered the sections in the corresponding positions. (Line 131, 197, 261, 406, 421, 502, 622, 773)

Comments 4: I have not provided detailed editorial revisions since I believe the journal staff will take care of it, but in cases where the wording affects the science I have given some minor revisions below:

line 197 - instead of 'biofuels as well as forage crops', revise to 'forage and biofuel crops'

line 230 - remove 'thus' since this sentence is not a continuation of any former information, but is the start of a new paragraph.

line 231 - remove 'however'; the second sentence agrees with the first, so there is no need for 'however'.

Response 4: We sincerely appreciate the valuable comments. We have made correction according to the reviewer’s comments in the corresponding positions.

Line 206: “For example, forage and biofuel crops generally have...”

Line 239: “Drought-induced flowering is a phenomenon more common in annual plants.”

Line 240: “Drought stress regulation of plant flowering has been less well studied in...”

Comments 5: The section starting at line 267 is a bit confusing. The information is there it seems, but is a bit disjointed, and in some cases it feels like the end of the thought is missing. Often, a flowering gene is referred to and then the effect of stress on the expression of the gene is described, but then there's no information about how flowering is affected. As a reader, I kept asking myself, and what about flowering? If the expression is affected by the stress, and it has X effect on the stress response, then what's the effect on flowering? Line 317 is a great example of what there could be more of - it's clear that the gene functions via drought to accelerate flowering. I would take this as an example of how things could be made clear in other sections. And if the answer is that you don't know what the effect of the stress is on flowering, then that point should be made clear.

Response 5: We sincerely appreciate the valuable comments. We have revised the manuscript in the corresponding positions.

We have deleted lines 284-286 because that portion of the text does not fit the logic of the subject matter.

Line 288-290: “Under long-day environmental conditions, water deprivation achieves drought-induced early flowering in Arabidopsis through ABA-dependent control of GI signaling that activates expression of the florigen genes FT and TSF.”

Line 326-329: “A nuclear factor-Y (NF-Y) transcription factor, ZmNF-YA3, which has the dual function of promoting maize flowering while increasing plant drought tolerance, but there is a lack of evidence on the specific mechanism of ZmNF-YA3 in drought-affected maize flower formation”

Line 330-333: “In citrus, CiNF-YA1 was also found to promote drought-induced flowering by forming a complex with CiNF-YB2 and CiNF-YC2 to activate CiFT expression, and overexpression of CiNF-YA1 in citrus increased plants drought-sensitive.”

Line 342-344: “Among them, CiFDα was induced by low temperature while CiFDβ was induced by drought stress. The regulatory mechanism of CiFDβ promoting drought-induced flowering is independent of FAC and interacts directly with AP1.”

Comments 6: line 279-281 - figure 1 is referenced here but WRKY44 is not in the figure. It should be made clear where exactly WRKY44 fits into the figure (if you can indicate a gene that it regulates, or is regulated by, in the figure then that would be sufficient.

Response 6: We sincerely appreciate the valuable comments. We have revised the manuscript in the corresponding positions.

Line 293-295: “It has been confirmed that GI-miR172 pathway was involved in drought escape by down-regulating WRKY44 (directly repress by miR172) (Figure 1).” 

At the same time, the specific position of WRKY44 in Figure 1 has been further clarified. (Line 272)

Comments 7: line 283 - again, the connection isn't quite clear here. You mention SVP and then mention the ABA pathway, but I think this should be elaborated upon because it isn't clear what the final consequence is. What happens, with respect to flowering, when the ABA pathway genes are inhibited? It just needs a small bit of extra information to complete the thought.

Response 7: We sincerely appreciate the valuable comments. We have revised the manuscript in the corresponding positions.

Line 295-299: “Several other flowering inhibition genes are also induced by drought stress. For example, water deficit induces the flowering repressor gene SVP, which represses the transcription of genes related to ABA catabolism, and increases ABA accumulation, which improves drought tolerance in Arabidopsis, but flowering is delayed.”

Comments 8: Line 285 - similar to above - you mention that the loss of FLC but the sentence, or idea, feels unfinished. loss of FLC leads to decreased drought tolerance, and what happens with flowering?

Response 8: We sincerely appreciate the valuable comments. We have revised the manuscript in the corresponding positions.

Line 299-301: “Similarly, FLC, a flowering suppressor gene, also plays a role in drought stress pathway, and the loss of FLC function leads to early flowering and decreased drought tolerance in Arabidopsis.”

Comments 9: Lines 749-750 - remove 'salt stress as well as well as poor nutrition stress' - this manuscript did not focus on those, so it doesn't seem like any conclusions should be made regarding them.

Response 9: We sincerely appreciate the valuable comments. We have made corrections at the appropriate positions in the manuscript.

Line 775-778: “Accordingly, this review outlines the current knowledge showing that abiotic stresses such as drought, low temperature and high temperature stress alter flowering time of plants.”

Comments on the Quality of English Language

Overall, the quality of English is fairly good, but I do suggest that the manuscript be checked over for small grammatical and structural revisions.

Response: We sincerely appreciate the valuable comments. We have thoroughly and meticulously scrutinized the manuscript and corrected any grammatical and formatting errors that existed.

Round 2

Reviewer 1 Report

Comments and Suggestions for Authors

My previous comments still stand, and I do not think this manuscript is suitable for publication. I have nothing more to comment.

Reviewer 2 Report

Comments and Suggestions for Authors

The authors have made the corrections I suggested. The article is acceptable.